# When Do Multi-Agent Systems Outperform? Analysing the Learning Efficiency of Agentic Systems

## Abstract

Reinforcement Learning (RL) has emerged as a crucial method for training or fine-tuning large language models (LLMs), enabling adaptive, task-specific optimizations through interactive feedback. Multi-Agent Reinforcement Learning (MARL), in particular, offers a promising avenue by decomposing complex tasks into specialized subtasks learned by distinct interacting agents, potentially enhancing the ability and efficiency of LLM systems. However, theoretical insights regarding when and why MARL outperforms Single-Agent RL (SARL) remain limited, creating uncertainty in selecting the appropriate RL framework. In this paper, we address this critical gap by rigorously analyzing the comparative sample efficiency of MARL and SARL within the context of LLM. Leveraging the Probably Approximately Correct (PAC) framework, we formally define SARL and MARL setups for LLMs, derive explicit sample complexity bounds, and systematically characterize how task decomposition and alignment influence learning efficiency. Our results demonstrate that MARL improves sample complexity when tasks naturally decompose into independent subtasks, whereas dependent subtasks diminish MARL's comparative advantage. Additionally, we introduce and analyze the concept of task alignment, quantifying the trade-offs when enforcing independent task decomposition despite potential misalignments. These theoretical insights clarify empirical inconsistencies and provide practical criteria for deploying MARL strategies effectively in complex LLM scenarios.

## 1 Introduction

Reinforcement Learning (RL) is a machine learning paradigm in which agents learn optimal decision-making through interaction with their environment, guided by feedback provided as rewards or penalties. Unlike supervised learning methods that rely exclusively on fixed labeled datasets, RL allows models to actively explore and adapt within dynamic environments, enabling continuous refinement of their behavior. Due to this capability for adaptation, RL has emerged as a foundational approach in modern artificial intelligence systems, notably in the training and fine-tuning of Large Language Models (LLMs) (Novikov et al., 2025; Jumper et al., 2021; Zhang et al., 2025b; Guo et al., 2025). A prominent RL approach used with LLMs is Reinforcement Learning from Human Feedback (RLHF), wherein models are trained to align closely with human preferences through iterative, interactive feedback provided by humans (Ziegler et al., 2019; Christiano et al., 2023; Dong et al., 2024). This interactive feedback mechanism naturally connects with the broader concept of **agentic systems**, in which LLMs function as active agents engaging with their environment, continually refining their policies based on received rewards and progressively optimizing their decision-making processes (Sutton et al., 1998; Silver et al., 2016).

Although agentic systems have a long and rich history, recent advances significantly extend traditional approaches by integrating sophisticated data-driven learning methods enabled by large-scale LLMs. Within this evolving paradigm, two primary frameworks have become prominent: Single-Agent Reinforcement Learning (SARL) and Multi-Agent Reinforcement Learning (MARL). SARL employs a unified agent that handles an entire complex task in an end-to-end fashion. By contrast, MARL explicitly decomposes complex tasks into multiple subtasks, each managed by specialized agents trained for distinct functionalities such as information retrieval, drafting textual content, con-

tent verification, or multi-stage reasoning and planning. This explicit task decomposition facilitates specialized optimization by individual agents, potentially enhancing scalability, efficiency, and overall task performance. Consequently, MARL frameworks have gained increasing attention for large-scale and sophisticated LLM applications (Tran et al., 2025), including retrieval-augmented generation (Nakano et al., 2022), tool-augmented language generation (Schick et al., 2023), collaborative content verification (Sun et al., 2024), and complex multi-step reasoning tasks (Yuan & Xie, 2025; Chen et al., 2024; Guo et al., 2024).

**Existing Gap.** Despite significant efforts and progress in MARL algorithm and system research, theoretical understanding and results for MARL remain limited (La Malfa et al., 2025). A fundamental question remains unresolved: **Under what conditions, if any, does MARL provide a clear and consistent advantage over SARL?** The existing literature shows conflicting evidence; some empirical studies demonstrate clear benefits of MARL (Ning & Xie, 2024), while others emphasize scenarios in which SARL is preferable (Feriani & Hossain, 2021). Currently, comprehensive theoretical frameworks rigorously clarifying these discrepancies and identifying precise conditions under which MARL consistently outperforms SARL are lacking. Addressing this theoretical gap is critical, as it will illuminate precisely when and why MARL should be adopted, guiding more informed and effective deployment of RL strategies in complex, resource-intensive tasks common in LLM training and fine-tuning.

**Challenges.** This paper addresses this critical gap by theoretically analyzing the sample efficiency of MARL and SARL, with a focus on LLM scenarios (sequence-to-sequence tasks). Two key challenges underpin this analysis. *First*, learning algorithms for both MARL and SARL are evolving rapidly; MARL, in particular, is still in its early development stages and lacking a universally accepted algorithmic standard. Therefore, our theoretical framework must remain agnostic to specific learning algorithms, focusing instead on general conditions and properties that may provide comparative advantages. *Second*, accurately characterizing **task decomposition** (the manner in which complex tasks are distributed among multiple agents) and quantifying **task alignment** (the degree to which MARL naturally aligns with the given task) are currently underdeveloped in theoretical research. Precisely modeling these concepts and understanding their impact on sample efficiency is crucial for rigorously comparing MARL and SARL. The insights will allow us to clearly identify conditions under which MARL provides a meaningful and consistent advantage over SARL.

**Contribution.** To address these challenges and our central research question, we adopt the Probably Approximately Correct (PAC) learning framework (Valiant, 1984), integrating perspectives of task decomposition and alignment based on the reward structure. Our results explicitly characterizes the sample complexities for MARL and SARL, identifying precise conditions under which MARL demonstrates superior sample efficiency. The main contributions are summarized as follows:

• We formulate MARL and SARL within the context of LLM tasks and perform rigorous PAC-based analyses of their sample complexities. Specifically, we derive theoretical learnability characterizations and establish sample complexity bounds for SARL (Theorem 4.1) and MARL under distinct task decomposition conditions—dependent subtasks (Theorem 4.2) and independent subtasks (Theorem 4.3). We show that when tasks are decomposable into independent subtasks, the MARL sample complexity is dominated by the most challenging subtask. Conversely, when subtasks exhibit dependencies, MARL complexity is influenced by the cumulative difficulty across subtasks. These foundational results provide critical insights towards understanding practical learning requirements and form a theoretical foundation for comparative analyses between MARL and SARL.

• Building on these foundational analyses, we rigorously examine how task decomposition shapes the relative effectiveness of MARL compared to SARL. We demonstrate explicitly that tasks decomposable into independent subtasks significantly favor MARL in terms of sample complexity (Proposition 4.4), whereas tasks with interdependent subtasks diminish MARL's advantages, yielding more nuanced comparative outcomes (Proposition 4.5). This theoretical characterization helps clarify previously inconsistent empirical observations and provides practical guidance for selecting between MARL and SARL based on inherent task characteristics.

• We further investigate the nuanced role of task alignment in MARL, examining the trade-off between alignment and MARL's sample efficiency advantage under forced independent task decomposition. We formally quantify task alignment as the discrepancy between MARL and SARL

reward functions. By deriving an explicit sample complexity bound for MARL in this misaligned scenario (Theorem 4.6), we clearly characterize the conditions under which MARL retains its sample efficiency advantage despite imperfect task alignment (Proposition 4.7). These findings provide guidance for deploying MARL strategies, offering a precise, quantifiable assessment of the alignment-decomposition trade-off.

## 2 RELATED WORK

**Agentic LLM System.** Recently, the integration of agentic systems with LLM has gained significant attention as a means to improve their capabilities in solving complex tasks (Zhang et al., 2025a; Schick et al., 2023; Yao et al., 2023; Zhang et al., 2024; Kazemnejad et al., 2024; Dai et al., 2024; Guo et al., 2025; Yu et al., 2025; Zheng et al., 2025). Unlike traditional models that generate outputs from fixed inputs, agentic LLMs actively interact with their environments, continuously adjusting their behavior based on dynamic feedback and reward signals. RL, especially RLHF, has become a dominant approach in developing these agentic LLM systems, allowing models to iteratively refine their outputs by learning from human-generated or environmental feedback (Christiano et al., 2023; Ouyang et al., 2022; Ziegler et al., 2019; Dong et al., 2024). Within the realm of agentic LLMs, two primary paradigms have emerged: SARL and MARL. While SARL involves a single model optimizing task performance independently, MARL decomposes complex tasks into subtasks, enabling specialized agents to collaborate or compete to achieve overall objectives (Tran et al., 2025; Yuan & Xie, 2025; Shinn et al., 2023). MARL has shown promise in empirical studies, particularly for tasks benefiting from specialization, parallelization, and cooperative reasoning (Sun et al., 2024; Liang et al., 2024; Guo et al., 2024). However, the critical question of precisely when and why MARL outperforms SARL remains largely unanswered. Existing work primarily relies on empirical demonstrations without offering comprehensive theoretical explanations or clearly defining the conditions for MARL's superiority (Ning & Xie, 2024). This paper addresses this critical gap by providing a formal theoretical analysis of the comparative sample efficiency between MARL and SARL in the context of LLM-based agentic systems.

**Theoretical Analysis of RL.** Another important direction related to this work is the theoretical investigation of RL, which has been extensively studied in the context of both single-agent and multi-agent systems (Shoham & Leyton-Brown, 2008; Zhang et al., 2025a). Theoretical research on RL typically focuses on objectives such as regret bounds (Bubeck & Cesa-Bianchi, 2012; He et al., 2021; Jin et al., 2019; Hu et al., 2023; Xiong et al., 2023), which measure the difference between the agent's performance and the optimal performance, and convergence properties, which describe how quickly an agent can learn an optimal policy (Azar et al., 2017; Sutton et al., 1998; Agarwal et al., 2020; Zhao et al., 2024). These works have provided fundamental insights into the dynamics of learning in RL environments and have been central to developing efficient RL algorithms. Additionally, value decomposition methods, which explore how complex tasks can be decomposed into subtasks to improve learning efficiency, have received significant attention (Koller & Parr, 1999; Kearns & Koller, 1999; Guestrin et al., 2002; Sunehag et al., 2017; Rashid et al., 2018; Son et al., 2019). Recent work has also extensively studied sample complexity within conventional RL frameworks (Hastie et al., 2009; Kearns & Singh, 2002; Dann et al., 2017; Zhan et al., 2022; Wagenmaker et al., 2022; Tirinzoni et al., 2022; 2023). However, existing analyses have primarily targeted standard RL settings rather than the specialized agentic scenarios typical of LLM training. Our work extends this body of theoretical research by explicitly analyzing sample complexity within the context of LLM-based agentic systems, placing a particular emphasis on task decomposition and alignment. Through this analysis, we provide precise conditions under which MARL achieves clear sample efficiency advantages over SARL, thereby advancing the theoretical understanding of RL in complex, agentic LLM scenarios.

## 3 PRELIMINARIES AND PROBLEM FORMULATION

In this section, we introduce the learning frameworks for SARL and MARL within the context of LLMs. We also briefly describe the PAC learning framework employed for analyzing sample complexity and explicitly state the key regularity assumptions underpinning our theoretical analysis.

**Notation.** We use lowercase letters to denote scalars and lower and uppercase boldface letters to denote vectors and matrices. For a vector $\boldsymbol{x}$, $\boldsymbol{x}[i]$ denotes the $i$-th coordinate. For two functions $f(x) \geq 0$ and $g(x) \geq 0$ defined on $x > 0$, we write $f(x) \lesssim g(x)$ if $f(x) \leq c \cdot g(x)$ for some absolute constant $c > 0$; we write $f(x) \gtrsim g(x)$ if $g(x) \lesssim f(x)$; we write $f(x) \asymp g(x)$ if $f(x) \lesssim g(x)$ and $g(x) \lesssim f(x)$. We denote $[k] := 1, \ldots, k$.

## 3.1 SINGLE-AGENT REINFORCEMENT LEARNING (SARL) FORMULATION

Let $\mathcal{T}$ denote the considered task and $\boldsymbol{x} \in \mathcal{X}$ denote a prompt drawn from a data distribution $\mathbb{D}$ (the input to the model). An LLM policy $\pi_{\boldsymbol{\theta}}$ generates a sequence $\boldsymbol{y} = (a_1, \ldots, a_T)$ from a finite vocabulary $\mathcal{V}$, conditioned on the evolving text state $\boldsymbol{s}_t = (\boldsymbol{x}, \boldsymbol{a}_{1:t-1})$. Transitions are deterministic: $\boldsymbol{s}_{t+1} = (\boldsymbol{s}_t, a_t)$. In the standard *sequence-level reward* setting (as in RLHF), a scalar reward $\mathrm{R}(\boldsymbol{x}, \boldsymbol{y}) \in \mathbb{R}$ is received after the full sequence is produced. The population objective is,

$$\mathrm{J}(\boldsymbol{\theta}) = \mathbb{E}_{\boldsymbol{x} \sim \mathbb{D}, \, \boldsymbol{y} \sim \pi_{\boldsymbol{\theta}}(\cdot \mid \boldsymbol{x})} \big[ \mathrm{R}(\boldsymbol{x}, \boldsymbol{y}) \big]$$

This may be viewed as a contextual bandit over sequences (no intermediate rewards) or an MDP with delayed reward. The policy class then can be defined as $\Pi = \{\pi_{\boldsymbol{\theta}} : \boldsymbol{\theta} \in \Theta\}$ with $\Theta \subset \mathbb{R}^d$. We note that the dimension $d$ corresponds the *effective dimension* of the trainable parameter subset for agent (e.g., LoRA rank or adapter dimension).

**Empirical optimization.** Given $n$ samples $\mathcal{S} = \{(\boldsymbol{x}_i, \boldsymbol{y}_i)\}_{i=1}^n$ from $\mathbb{D}$ (e.g., rollouts or logged data), the empirical objective is given by,

$$\widehat{\mathrm{J}}_n(\boldsymbol{\theta}) = \frac{1}{n} \sum_{i=1}^n \mathrm{R}(\boldsymbol{x}_i, \boldsymbol{y}_i), \qquad \widehat{\boldsymbol{\theta}} \in \arg\max_{\boldsymbol{\theta}} \widehat{\mathrm{J}}_n(\boldsymbol{\theta}).$$

Typical reward functions $\mathrm{R}(\boldsymbol{x}, \boldsymbol{y})$ encountered in practice include human preference scores (Ziegler et al., 2019; Christiano et al., 2023), correctness indicators for question-answering tasks (Nakano et al., 2022), or metrics that evaluate coherence and relevance in text generation (Zhang et al., 2020). The optimization is commonly solved through standard gradient-based methods, including gradient descent, stochastic gradient descent, or advanced policy gradient methods such as PPO and GRPO (Schulman et al., 2017; Rafailov et al., 2024).

## 3.2 MULTI-AGENT REINFORCEMENT LEARNING (MARL) FORMULATION

Our aim is to compare SARL and MARL *on the same task and reward*. To make the comparison transparent and align with the LLM context, we adopt a *segment factorization* of the output: the response $\boldsymbol{y}$ is partitioned into $K$ contiguous segments $\boldsymbol{y}^{(1)}, \ldots, \boldsymbol{y}^{(K)}$ such that

$$\boldsymbol{y} = \mathrm{concat}\big(\boldsymbol{y}^{(1)}, \ldots, \boldsymbol{y}^{(K)}\big), \qquad \boldsymbol{y}^{(<i)} := \mathrm{concat}\big(\boldsymbol{y}^{(1)}, \ldots, \boldsymbol{y}^{(i-1)}\big).$$

**Specialized, turn-taking policies.** MARL replaces the single monolithic policy with $K$ *specialized* policies, one per segment, executed sequentially (turn-taking):

$$\pi_{\boldsymbol{\theta}_i} : (\boldsymbol{x}, \boldsymbol{y}^{(<i)}) \longmapsto \pi_{\boldsymbol{\theta}_i}(\cdot \mid \boldsymbol{x}, \boldsymbol{y}^{(<i)}) \in \Delta(\mathcal{V}^{T_i}), \quad i = 1, \ldots, K,$$

with sequential sampling:

$$\boldsymbol{y}^{(1)} \sim \pi_{\boldsymbol{\theta}_1}(\cdot \mid \boldsymbol{x}), \qquad \boldsymbol{y}^{(i)} \sim \pi_{\boldsymbol{\theta}_i}(\cdot \mid \boldsymbol{x}, \boldsymbol{y}^{(<i)}), \, i = 2{:}K, \qquad \boldsymbol{y} = \mathrm{concat}\big(\boldsymbol{y}^{(1)}, \ldots, \boldsymbol{y}^{(K)}\big).$$

We denote the MARL parameter as $\overline{\boldsymbol{\theta}} = (\boldsymbol{\theta}_1, \ldots, \boldsymbol{\theta}_K)$ with $\boldsymbol{\theta}_i \in \Theta_i \subset \mathbb{R}^{d_i}$. Then, we can write the joint policy as $\pi_{\overline{\boldsymbol{\theta}}} = (\pi_{\boldsymbol{\theta}_1}, \ldots, \pi_{\boldsymbol{\theta}_K})$. For simplicity, we assume that all agents condition on the same global prompt $\mathbf{x}$ together with the conversation history available up to their turn. This setup mirrors common multi-agent LLM pipelines (e.g., planner $\to$ solver $\to$ verifier), where later agents observe earlier outputs and therefore act under *monotonically increasing* context rather than identical observations. Importantly, none of our later theoretical results depend on this particular modelling choice. If one prefers to model agent-specific partial observations, the analysis extends directly with modest modification.

**Task decomposition.** To rigorously analyze how task decomposition impacts the performance of MARL, we assume that the original unified reward function $R(\boldsymbol{x}, \boldsymbol{y})$ can be equivalently represented in two distinct decomposed forms, each capturing different assumptions about subtask dependency:

$$\overline{R}_{\mathrm{dep}}(\boldsymbol{x}, \boldsymbol{y}) = \frac{1}{K} \sum_{i=1}^{K} r_i\big(\boldsymbol{x}, \boldsymbol{y}^{(i)}, \boldsymbol{y}^{(<i)}\big), \tag{3.1}$$

$$\overline{R}_{\mathrm{indep}}(\boldsymbol{x}, \boldsymbol{y}) = \frac{1}{K} \sum_{i=1}^{K} r_i\big(\boldsymbol{x}, \boldsymbol{y}^{(i)}\big), \tag{3.2}$$

where $r_i(\cdot)$ is the reward function for the $i$-th agent. These two formulations capture a spectrum of practical task decomposition scenarios, ranging from fully independent segments (idealized settings) to dependent subtasks that capture realistic interdependencies. Specifically, equation 3.1 explicitly models dependent subtasks, incorporating forward coherence by conditioning each segment's reward on preceding segments $\boldsymbol{y}^{(<i)}$. In contrast, equation 3.2 represents an idealized scenario where subtasks are fully independent, with each segment's reward depending solely on the corresponding segment. These formulations enable us to rigorously analyze the influence of task dependency on MARL's sample complexity, which we investigate systematically in subsequent sections.

The MARL learning objective is thus defined as:

$$J_{\mathrm{MARL}}(\pi_{\overline{\boldsymbol{\theta}}}) = \mathbb{E}_{\boldsymbol{x} \sim \mathbb{D}, \, \boldsymbol{y} \sim \pi_{\overline{\boldsymbol{\theta}}}(\cdot|\boldsymbol{x})} \big[\overline{R}_{\mathrm{dep/indep}}(\mathbf{x}, \mathbf{y})\big]$$

**Empirical Optimization.** Using dataset $\mathcal{S} = \{(\boldsymbol{x}_j, \boldsymbol{y}_j)\}_{j=1}^{n}$, the empirical MARL objective is:

$$\widehat{J}_{\mathrm{MARL},n}(\pi_{\overline{\boldsymbol{\theta}}}) = \frac{1}{n} \sum_{j=1}^{n} \overline{R}(\boldsymbol{x}_j, \boldsymbol{y}_j), \qquad \widehat{\pi}_{\overline{\boldsymbol{\theta}}} \in \arg\max_{\pi} \widehat{J}_{\mathrm{MARL},n}(\pi_{\overline{\boldsymbol{\theta}}}).$$

### 3.3 SAMPLE COMPLEXITY UNDER THE PAC FRAMEWORK

To rigorously compare the sample efficiency between SARL and MARL in the LLM context, we employ the PAC learning framework (we present a more detailed and complete introduction in Appendix C for completeness purpose). Within this framework, the concept of sample complexity refers to the number of samples (or equivalently, environment interactions) required to learn an $\varepsilon$-optimal policy with high probability (at least $1 - \delta$). Formally, the PAC learnability within our context can be defined as follows:

**Definition 1** (PAC Learnability). *Consider a policy class $\Pi$ and define the expected reward associated with a policy $\pi \in \Pi$ as: $J(\pi) := \mathbb{E}_{\boldsymbol{x} \sim \mathbb{D}, \, \boldsymbol{y} \sim \pi(\cdot|\boldsymbol{x})}[R(\boldsymbol{x}, \boldsymbol{y})]$. We say the policy class $\Pi$ is PAC-learnable if, for any accuracy parameter $\varepsilon > 0$ and confidence parameter $\delta \in (0, 1)$, there exists a finite sample complexity $N(\varepsilon, \delta)$ such that, for all $n \geq N(\varepsilon, \delta)$, any empirical maximizer*

$$\widehat{\pi} \in \arg\max_{\pi \in \Pi} \widehat{J}_n(\pi), \quad \text{where} \quad \widehat{J}_n(\pi) = \frac{1}{n} \sum_{i=1}^{n} R(\boldsymbol{x}_i, \boldsymbol{y}_i), \quad (\boldsymbol{x}_i, \boldsymbol{y}_i) \sim \mathbb{D}, \quad \text{satisfies:}$$

$$\Pr\left[J(\widehat{\pi}) \geq \sup_{\pi \in \Pi} J(\pi) - \varepsilon\right] \geq 1 - \delta.$$

**Comparison Framework.** This PAC formulation forms the theoretical foundation for systematically comparing the sample complexities of SARL and MARL. Specifically, our analysis involves the following key steps:

1. **Derivation of explicit PAC bounds:** We explicitly derive the PAC sample complexity bounds necessary to achieve an $\varepsilon$-optimal policy for both SARL and MARL.

2. **Comparative analysis of sample complexity bounds:** Using these derived bounds, we rigorously examine the relative efficiency between SARL and MARL. In particular, we highlight how *task decomposition* (the degree to which a task can naturally be segmented among multiple agents) influences the relative sample complexity of the two frameworks.

3. **Impact of task alignment:** Finally, we investigate the role of *task alignment* within MARL, explicitly characterizing how alignment between subtask structures and the MARL framework impacts overall learning efficiency.

Ultimately, this structured approach enables us to formally identify and clearly articulate the conditions under which MARL yields statistically significant advantages over SARL (and vice versa), providing precise theoretical criteria to guide the selection of appropriate reinforcement learning frameworks in practical LLM-based scenarios.

### 3.3.1 REGULARIZATION AND ASSUMPTIONS

Next, we outline key regularity assumptions underpinning our analysis, ensuring theoretical tractability and alignment with practical LLM–RL scenarios. Modern RL practices for LLM operate under the following structural and regularizing constraints: (i) Generated text sequences are bounded by a finite vocabulary and a fixed maximum length $T_{\max}$; (ii) Policies are trained with explicit capacity controls, such as weight decay, norm or gradient clipping, KL-divergence constraints around reference policies, or low-rank/adapter parameterizations, resulting in a compact parameter set $\Theta$; (iii) Common neural network architectures induce Lipschitz continuity, ensuring smoothness and bounded variation in action distributions and sequence-level values.

These conditions guarantee finite metric entropy and covering numbers for the induced policy class, enabling rigorous PAC-style sample complexity analysis. Based on these practical considerations, we formally state our assumptions as follows:

**Assumption 1** (Bounded Reward). *The reward functions satisfy $0 \leq \mathrm{R}(\boldsymbol{x}, \boldsymbol{y}) \leq 1$ for SARL and $0 \leq \mathrm{r}_i(\boldsymbol{x}, \boldsymbol{y}) \leq 1$ for each MARL agent $i$.*

**Assumption 2** (Compact Parameter Sets). *The parameter spaces are compact: $\Theta \subseteq \mathbb{R}^d$ with $\|\boldsymbol{\theta}\|_2 \leq B$ for SARL, and $\Theta_i \subseteq \mathbb{R}^{d_i}$ with $\|\boldsymbol{\theta}_i\|_2 \leq B_i$ for each MARL agent $i$.*

**Assumption 3** (Lipschitz parameterization). *For both SARL and MARL policies, there exists a finite constant $L_{\mathrm{step}} > 0$ such that, for any text state $\mathbf{s}$ and any parameters $\boldsymbol{\theta}, \boldsymbol{\theta}' \in \Theta$, the policies satisfy:*

$$\left\|\pi_{\boldsymbol{\theta}}(\cdot \mid \mathbf{s}) - \pi_{\boldsymbol{\theta}'}(\cdot \mid \mathbf{s})\right\|_1 \leq L_{\mathrm{step}} \|\boldsymbol{\theta} - \boldsymbol{\theta}'\|_2.$$

**Assumption 4** (Finite Horizon). *The sequence generation horizons are bounded by finite constants $T_{\max}$ (SARL) and $T_{\max,i}$ (each MARL agent $i$).*

Detailed justification and further discussion are provided in Appendix B.

## 4 MAIN RESULTS

In this section, we present the primary theoretical findings of our analysis. We first provide the PAC sample complexity bounds for both SARL and MARL under different task decomposition scenarios. Subsequently, we explicitly characterize the relative sample efficiency between SARL and MARL. Finally, we analyse the impact of task misalignment on MARL's efficiency. Detailed proofs for these results are provided in the Appendix.

### 4.1 SAMPLE COMPLEXITY BOUNDS FOR SARL AND MARL

We begin by establishing the PAC sample complexity bound for the SARL setting:

**Theorem 4.1** (PAC sample complexity for SARL). *Under Assumptions 1–4, the SARL framework is PAC-learnable with sample complexity:*

$$\mathrm{N}_{\mathrm{SARL}}(\varepsilon, \delta) = \mathcal{O}\left(\frac{d \log(L_{seq}B/\varepsilon) + \log(1/\delta)}{\varepsilon^2}\right), \quad L_{\mathrm{seq}} = T_{\max}L_{\mathrm{step}}.$$

Theorem 4.1 demonstrates that the sample complexity of SARL scales with the (effective) dimension $d$, sequence length $L_{\mathrm{seq}}$, and parameter space radius $B$. This provides foundational insights into the key trade-offs that influence SARL's learning efficiency.

Next, we provide results for MARL under dependent and independent task decompositions:

**Theorem 4.2** (PAC Sample Complexity for MARL: Dependent Sub-task). *Consider a MARL system of $K$ agents whose task structure follows equation 3.1, i.e., tasks decompose into dependent subtasks. Under Assumptions 1–4, MARL is PAC-learnable with sample complexity:*

$$N_{\text{MARL}}(\varepsilon, \delta) = \mathcal{O}\left(\frac{\sum_{i=1}^{K} d_i \log(L_{seq,i} B_i/\varepsilon) + \log(K/\delta)}{(\varepsilon/K)^2}\right), \quad L_{\text{seq},i} = T_{\max,i} L_{\text{step}}.$$

**Theorem 4.3** (PAC Sample Complexity for MARL: Independent Sub-task). *Consider a MARL system of $K$ agents whose task structure follows equation 3.2, i.e., tasks decompose into independent subtasks. Under Assumptions 1–4, MARL is PAC-learnable with sample complexity:*

$$N_{\text{MARL}}(\varepsilon, \delta) = \mathcal{O}\left(\frac{\widetilde{d} \log(\gamma/\varepsilon) + \log(1/\delta)}{\varepsilon^2}\right),$$

*where $\widetilde{d} = \max\{d_1, ..., d_K\}$ and $\gamma = \max\{L_{\text{seq},1} B_1, ..., L_{\text{seq},K} B_K\}$ with $L_{\text{seq},i} = T_{\max,i} L_{\text{step}}$.*

Theorems 4.2 and 4.3 characterize the PAC sample complexity bounds for MARL under dependent and independent task decomposition scenarios, respectively. Specifically, Theorem 4.2 shows that when subtasks exhibit interdependencies, the overall sample complexity increases, scaling roughly with the cumulative complexities of all component tasks. Moreover, the additional $K^2$ factor reflects the fact that dependencies introduce *error propagation*: estimation errors in early agents can amplify through subsequent agents, leading to a quadratic penalty in the worst case. This indicates that task interdependencies impose an additional learning burden due to coordination and coherence requirements among subtasks. In contrast, Theorem 4.3 demonstrates that if tasks decompose into fully independent subtasks, the total sample complexity is dominated solely by the most challenging individual subtask. These results underscore the critical role played by task decomposition structure, showing that MARL achieves greater efficiency when subtasks are independent, but can incur higher complexity when significant interdependencies exist.

## 4.2 RELATIVE SAMPLE EFFICIENCY

Leveraging the results above, we can now compare the relative sample efficiency between SARL and MARL under different task decompositions.

**Proposition 4.4.** *Under assumptions of Theorems 4.1 and 4.3, MARL consistently achieves equal or superior sample efficiency compared to SARL:*

$$N_{\text{MARL}}(\varepsilon, \delta) \lesssim N_{\text{SARL}}(\varepsilon, \delta).$$

*In particular, when the task decomposes into $K$ homogeneous independent segments (i.e., $\widetilde{d} = d/K$ and $T_{\max,i} = T_{\max}/K$ for all segments), the efficiency gain becomes explicit:*

$$\frac{N_{\text{MARL}}(\varepsilon, \delta)}{N_{\text{SARL}}(\varepsilon, \delta)} \lesssim \frac{1}{K}.$$

Proposition 4.4 provides a formal characterization of MARL's efficiency advantage over SARL under independent task decomposition. Specifically, it establishes that MARL consistently achieves equal or superior sample complexity compared to SARL. Moreover, in scenarios involving homogeneous, independent subtasks, MARL exhibits significant efficiency gains as the number of agents grows, effectively reducing the overall sample complexity by partitioning the global task into smaller, independently learnable components. This result underscores how leveraging task independence can substantially enhance MARL's learning efficiency relative to SARL.

**Proposition 4.5.** *Under the assumptions of Theorems 4.1 and 4.2, the relative sample complexity between MARL and SARL satisfies:*

$$\frac{N_{\text{MARL}}(\varepsilon, \delta)}{N_{\text{SARL}}(\varepsilon, \delta)} \lesssim K^2 \mathcal{A}(\varepsilon)\mathcal{C}(\varepsilon, \delta), \quad where,$$

$$\mathcal{A}(\varepsilon) = \sum_{i=1}^{K} \frac{d_i \, \log(L_{seq,i}B_i/\varepsilon)}{d \, \log(L_{seq}B/\varepsilon)}, \quad \mathcal{C}(\varepsilon,\delta) = \frac{1 + \dfrac{\log(K/\delta)}{\sum_i d_i \log(L_{\mathrm{seq},i}B_i/\varepsilon)}}{1 + \dfrac{\log(1/\delta)}{d \log(L_{\mathrm{seq}}B/\varepsilon)}}.$$

Proposition 4.5 explicitly characterizes the relative sample efficiency between MARL and SARL in settings involving dependent subtasks. The factors influencing this relative efficiency are clearly encapsulated by the decomposition ratio factor $\mathcal{A}(\varepsilon)$, which quantifies how closely the subtasks align with a natural decomposition, and the confidence correction factor $\mathcal{C}(\varepsilon,\delta)$, which accounts for statistical confidence adjustments. The result highlights that subtask dependencies introduce coordination overhead, potentially diminishing MARL's inherent advantage. Specifically, the sample complexity ratio depends critically on the factors $K^2$, $\mathcal{A}(\varepsilon)$, and $\mathcal{C}(\varepsilon,\delta)$, underscoring that MARL typically outperforms SARL under independent decompositions but may incur substantial efficiency losses as subtask dependencies grow.

In particular, considering practical scenarios involving large-scale models—where the dimensions of model parameters ($d$ and $\sum_i d_i$) typically dominate other terms—we can approximate the confidence correction as $\mathcal{C}(\varepsilon,\delta) \approx 1$ and the alignment factor as $\mathcal{A}(\varepsilon) \approx \sum_i d_i/d$. Under these conditions, MARL maintains a clear sample complexity advantage over SARL if and only if:

$$K^2 \sum_i d_i/d \lesssim 1.$$

This condition intuitively indicates that MARL remains advantageous if subtasks can be effectively learned by specialized models that individually require fewer parameters compared to the single unified SARL model. Consequently, the practical efficiency of MARL relies critically on utilizing smaller, specialized sub-models whose combined parameter complexity is significantly lower than that of the unified SARL counterpart.

### 4.3 Cost of Task Misalignment

Earlier results have demonstrated that MARL holds significant advantages over SARL when a task naturally decomposes into independent subtasks and the associated reward structure aligns appropriately. However, in practice, tasks often do not perfectly align with independent decompositions. A critical practical question thus arises: *Can we still exploit MARL's benefits by enforcing independent task decompositions, despite potential misalignment with the original task structure?* We rigorously analyze this trade-off in this section.

**Quantifying Task Misalignment.** To explicitly measure misalignment, we introduce a *task alignment factor* $\alpha$, defined as the maximum discrepancy between the MARL reward and the unified SARL reward. Formally, let $\overline{\mathrm{R}}(\boldsymbol{x},\boldsymbol{y})$ denote the MARL-decomposed reward and $\mathrm{R}(\boldsymbol{x},\boldsymbol{y})$ the SARL reward. The alignment factor $\alpha$ is:

$$\alpha := \sup_{\mathbf{x},\mathbf{y}}, \left|\overline{\mathrm{R}}(\mathbf{x},\mathbf{y}) - \mathrm{R}(\mathbf{x},\mathbf{y})\right|. \tag{4.1}$$

The quantity $\alpha$ measures the worst-case misalignment between the true reward and its independent decomposition. A small value indicates that the subtasks faithfully preserve the structure of the original problem, whereas a large value reflects substantial distortion introduced by the decomposition.

The following theorem quantifies the impact of task misalignment on MARL's sample complexity:

**Theorem 4.6** (PAC Sample Complexity for MARL). *Consider a MARL system with $K$ agents whose subtasks are independently decomposed (Eq. 3.2), but exhibit task alignment $\alpha$ as defined above. Under Assumptions 1–4 and given that $\varepsilon > 2\alpha$, MARL is PAC-learnable with sample complexity:*

$$\mathrm{N}_{\mathrm{MARL}}(\varepsilon,\delta) \;=\; \mathcal{O}\left( \frac{\widetilde{d} \, \log\!\left(\dfrac{K\gamma}{\varepsilon - 2\alpha}\right) + \log(1/\delta)}{(\varepsilon - 2\alpha)^2} \right),$$

*where $\widetilde{d} = \max\{d_1,\ldots,d_K\}$ and $\gamma = \max\{L_{\mathrm{seq},1}B_1,\ldots,L_{\mathrm{seq},K}B_K\}$.*

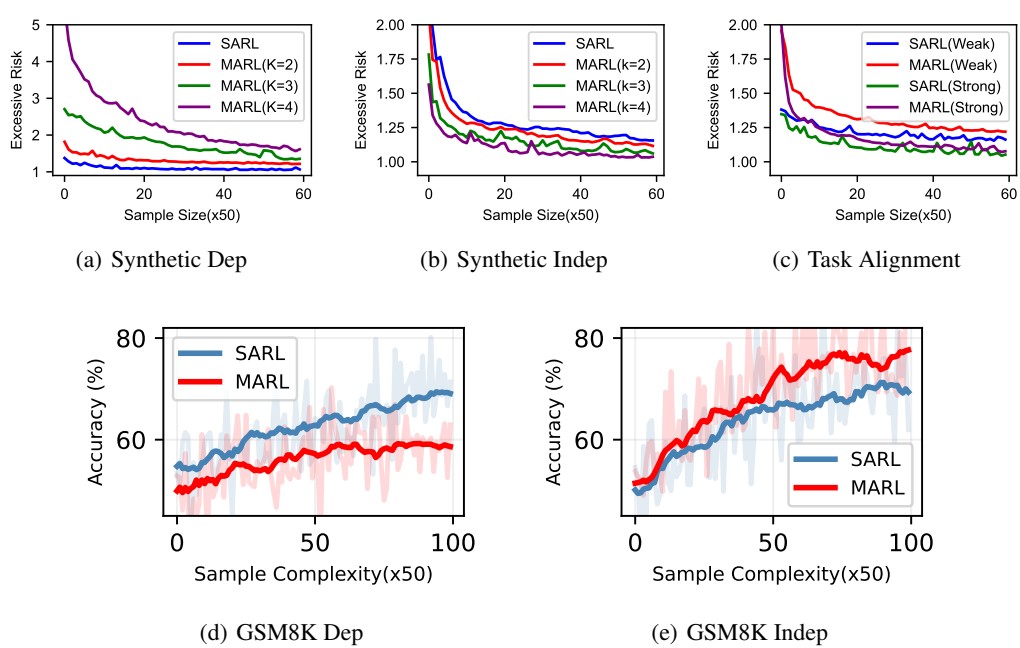

Figure 1: Empirical comparison of SARL and MARL learning efficiency under (a) dependent and (b) independent task decompositions using synthetic tasks, and under (d) dependent and (e) independent decompositions on GSM8K. (c) illustrates the effect of task alignment on the relative learning efficiency of SARL and MARL, where "strong" indicates high alignment and "weak" indicates low alignment.

Theorem 4.6 quantifies how misalignment impacts the feasibility and efficiency of MARL. Specifically, it shows that substantial misalignment ($\alpha$) relative to the desired accuracy ($\varepsilon$) can render MARL impractical. Conversely, when misalignment is within a manageable range, it can be mitigated through an increase in sample size.

Building upon this result, we derive a precise condition that determines when MARL maintains its advantage over SARL despite task misalignment:

**Proposition 4.7** (Condition for MARL Advantage under Imperfect Alignment). *Under the conditions specified in Theorems 4.1 and 4.6, MARL achieves superior sample efficiency over SARL, namely:*

$$\mathrm{N}_{\mathrm{MARL}}(\varepsilon, \delta) \lesssim \mathrm{N}_{\mathrm{SARL}}(\varepsilon, \delta), \quad if:$$

$$\kappa_d \kappa_\ell \le \left(1 - \frac{2\alpha}{\varepsilon}\right)^2, \quad where \quad \kappa_d := \frac{\widetilde{d}}{d}, \quad \kappa_\ell := \frac{\log\left(\frac{K\gamma}{\varepsilon - 2\alpha}\right)}{\log\left(\frac{L_{\mathrm{seq}} B}{\varepsilon}\right)}.$$

Proposition 4.7 identifies the conditions under which MARL retains sample efficiency advantages over SARL despite imperfect task decomposition. Thus, it provides a quantitative measure of the alignment-decomposition trade-off, serving as practical guidance for deploying MARL strategies.

### 4.4 EMPIRICAL STUDY

We conduct an empirical evaluation to validate our theoretical results, comparing the performance and sample efficiency of SARL and MARL across varying task decomposition scenarios.

**Experimental Setup.** We consider a synthetic sequence-to-sequence regression task generated under two distinct scenarios representing independent and dependent subtasks:

$$
y_i = \begin{cases} \mathbf{w}_i^\top \mathbf{x}_i + \xi_i, & \text{(independent subtasks)}, \\ \mathbf{w}_i^\top \mathbf{x}_i + \lambda \cdot \text{average}(y^{(<i)}) + \xi_i, & \text{(dependent subtasks)}, \end{cases} \qquad \xi_i \sim \mathbb{N}(0, \sigma^2)
$$

The independent scenario generates subtasks without interdependencies, while the dependent scenario explicitly incorporates dependencies among subtasks through the average of previous outputs. In both scenarios, we compare two configurations: (i) a single-agent system (SARL) employing a unified parameter space, and (ii) a multi-agent system (MARL) composed of sequentially trained specialized agents with distinct parameter spaces. The feature vector $\mathbf{x}_i$ drawn from Gaussian distributions and $\sigma^2 = 1$. Each experiment is conducted over five independent trials, and the averaged results are reported. We additionally evaluate on the real-world GSM8K math reasoning dataset (Cobbe et al., 2021). Independent and dependent subtasks are created either by concatenating multiple questions or by decomposing the solution process into "thinking → solving." For the synthetic experiments, each agent is implemented as a one-layer MLP, while for GSM8K we use `Qwen2.5-1.5B-Instruct` as the backbone model. Further

**Results.** Figure 1 presents our empirical results, which closely match the theoretical predictions. In the independent-subtask setting, MARL achieves markedly better sample efficiency than SARL, while in the dependent-subtask setting, SARL consistently outperforms MARL due to error propagation across agents. The contrast between the two regimes becomes more pronounced as the number of agents $K$ increases. Moreover, we observe that under strong task alignment $\lambda = 0.1$ (i.e.,small $\alpha$), MARL performs comparably to SARL, whereas misaligned decompositions $\lambda = 1$ (large $\alpha$) lead to the expected degradation in MARL performance.

## 5 CONCLUDING DISCUSSION

In this paper, we presented a theoretical analysis comparing the sample efficiency of MARL and SARL within LLM contexts. Leveraging the PAC learning framework, we characterized the sample complexity for both MARL and SARL and identified precise conditions under which MARL achieves superior sample efficiency. Our results demonstrate that MARL significantly reduces sample complexity when tasks naturally decompose into independent subtasks. Conversely, task dependencies diminish MARL's advantages, highlighting critical trade-offs in learning efficiency. Furthermore, we introduced and explicitly quantified the notion of task alignment to rigorously evaluate the consequences of enforcing independent task decompositions, even at the risk of misalignment with the original task structure. These theoretical insights reconcile empirical findings and provide practical guidelines for selecting reinforcement learning frameworks tailored to complex, agentic LLM applications. Consequently, our work bridges a critical gap between empirical practice and theoretical foundations, fostering informed and efficient reinforcement learning deployment.

**Limitations and Future Work.** Our study has several limitations that open promising avenues for future research. First, our theoretical framework deliberately abstracts away algorithm-specific optimization details to maintain generality. Future investigations could integrate specific policy-gradient algorithms—such as PPO, GRPO, or other widely-used optimization methods—to yield more precise, algorithm-dependent insights and actionable guidelines. Second, we focused primarily on idealized task decomposition scenarios: fully independent or fully dependent subtasks. Exploring intermediate scenarios involving partially dependent subtasks could yield deeper theoretical insights and offer more nuanced practical recommendations. Addressing these limitations will further strengthen the theoretical understanding of MARL and SARL, ultimately enhancing the applicability of reinforcement learning methods within complex LLM environments. Our analysis assumes i.i.d. training samples drawn from a fixed distribution, consistent with common LLM-based RL settings (e.g., RLHF, offline RLHF) where prompts are independently sampled and rewards are assigned per rollout. In fully interactive RL or MARL, trajectories can exhibit temporal dependence. Extending the theory to this regime would require replacing the i.i.d. assumption with mixing-time or martingale-based concentration tools (e.g., uniform convergence under $\beta$-mixing), introducing additional dependence on the mixing properties of the underlying Markov process.

ETHICS STATEMENT

We have carefully reviewed the ICLR Code of Ethics and confirm that our work does not raise any significant ethical concerns. The research is purely theoretical in nature and does not involve human subjects, personal or sensitive data, or applications that could pose potential societal harm. Our methodology adheres to established standards of fairness, transparency, and scientific integrity. All results are derived from rigorous analysis under clearly stated assumptions, and the contributions are intended solely to advance the understanding of reinforcement learning in large language models.

REPRODUCIBILITY STATEMENT

This paper is theoretical in nature, and all assumptions used in the analysis are explicitly stated in the main text. Full proofs and detailed derivations of our results are provided in the appendix to ensure clarity and verifiability. While the primary contributions are analytical, we also include complementary empirical demonstrations using synthetically generated data, for which we provide complete descriptions and configurations. The paper therefore discloses all necessary information—assumptions, theoretical setup, and data generation process—required to reproduce our results and verify the validity of the conclusions.

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

## THE USE OF LARGE LANGUAGE MODELS (LLMS)

We used LLMs only as a general-purpose writing assistant to aid in grammar checking and polishing the writing. The LLM did not contribute to research ideas, experiment design, theoretical analysis, or result interpretation.

## A    CURRENT PARADIGM OF RL TRAINING FOR LLMS

Reinforcement Learning (RL) has emerged as a key approach for training and fine-tuning Large Language Models (LLMs), particularly in scenarios where models must generate accurate, contextually coherent, and human-aligned responses. In contrast to traditional supervised learning, RL enables models to iteratively improve their performance through feedback and interactive experiences.

### A.1    COMMON RL OBJECTIVES FOR LLMS

In typical RL-based training frameworks for LLMs, the primary objective is maximizing a sequence-level expected reward. Formally, given a prompt $\boldsymbol{x} \sim \mathcal{D}$ and a generated sequence of tokens $\boldsymbol{y} = (a_1, \ldots, a_T)$ drawn from the policy $\pi_{\boldsymbol{\theta}}$, the learning objective can be expressed as:

$$J(\boldsymbol{\theta}) = \mathbb{E}_{\boldsymbol{x} \sim \mathcal{D},\, \boldsymbol{y} \sim \pi_{\boldsymbol{\theta}}(\cdot|\boldsymbol{x})}[R(\boldsymbol{x}, \boldsymbol{y})], \tag{A.1}$$

where $R(\boldsymbol{x}, \boldsymbol{y}) \in [0, 1]$ denotes a scalar reward evaluated after the sequence is produced.

Commonly used reward functions in practice include:

- **Human preference scores**: Human annotators evaluate generated outputs for factors like coherence, accuracy, and relevance.

- **Task-specific correctness indicators**: Binary or continuous evaluations of task performance such as factual correctness or logical accuracy.

- **Automated evaluation metrics**: Metrics such as BLEU, ROUGE, and BERTScore measure linguistic quality, coherence, and semantic similarity to reference texts.

### A.2    OPTIMIZATION ALGORITHMS

Contemporary RL training for LLMs predominantly employs policy gradient methods, often incorporating regularization techniques or trust-region constraints to ensure stable training and controlled updates. Two prominent methods are Proximal Policy Optimization (PPO) and Groupwise Relative Policy Optimization (GRPO):

**Proximal Policy Optimization (PPO)**    PPO optimizes a clipped surrogate objective to stabilize updates:

$$\begin{aligned} L^{\mathrm{PPO}}(\boldsymbol{\theta}) = \mathbb{E}_{\boldsymbol{x}, \boldsymbol{y} \sim \pi_{\boldsymbol{\theta}^-}} &\Big[ \min\big(r_{\boldsymbol{\theta}}(\boldsymbol{x}, \boldsymbol{y}) A^-,\, \mathrm{clip}(r_{\boldsymbol{\theta}}(\boldsymbol{x}, \boldsymbol{y}), 1 - \epsilon, 1 + \epsilon) A^-\big) \Big] \\ &- \beta\, \mathbb{E}_{\boldsymbol{x}}[D_{\mathrm{KL}}(\pi_{\boldsymbol{\theta}} \| \pi_{\boldsymbol{\theta}^-})], \end{aligned} \tag{A.2}$$

where:

- $r_{\boldsymbol{\theta}}(\boldsymbol{x}, \boldsymbol{y}) = \frac{\pi_{\boldsymbol{\theta}}(\boldsymbol{y}|\boldsymbol{x})}{\pi_{\boldsymbol{\theta}^-}(\boldsymbol{y}|\boldsymbol{x})}$ denotes the importance ratio relative to a previous policy $\pi_{\boldsymbol{\theta}^-}$.

- $A^-(\boldsymbol{x}, \boldsymbol{y}) = R(\boldsymbol{x}, \boldsymbol{y}) - V^-(\boldsymbol{x})$ is the advantage estimate.

- $\epsilon > 0$ controls the update step size, and $\beta \geq 0$ regularizes updates via KL divergence constraints.

This method is widely applied due to its stability and efficiency in LLM fine-tuning.

**Groupwise Relative Policy Optimization (GRPO)**   GRPO uses group-based reward comparisons to enhance training stability and reduce variance:

$$L^{\text{GRPO}}(\boldsymbol{\theta}) = \mathbb{E}_{\boldsymbol{x}\sim\mathcal{D},\,\boldsymbol{y}^{1:G}\sim(\pi_{\boldsymbol{\theta}^-})^{\otimes G}} \left[ \frac{1}{G} \sum_{g=1}^{G} w(\boldsymbol{x}, \boldsymbol{y}^{1:G}, g)\, r_{\boldsymbol{\theta}}(\boldsymbol{x}, \boldsymbol{y}^{(g)}) \right]$$
$$- \beta\, \mathbb{E}_{\boldsymbol{x}}[D_{\text{KL}}(\pi_{\boldsymbol{\theta}} \| \pi_{\boldsymbol{\theta}^-})], \tag{A.3}$$

where:

- $w(\boldsymbol{x}, \boldsymbol{y}^{1:G}, g)$ represents a group-relative reward, typically constructed to be centered, reducing estimator variance.
- $G$ denotes the number of completions sampled per prompt, enabling inter-group reward comparison.

GRPO improves training efficiency and reliability, especially when evaluating subtler quality differences between multiple generated sequences.

**Summary of Current RL Paradigm for LLMs**   In summary, current LLM RL frameworks share key characteristics:

1. Optimization of a sequence-level reward function reflecting task-specific performance, correctness, or human preferences.
2. Autoregressive policies generating sequences token-by-token, enabling sequential decision-making.
3. KL-regularized objectives or trust-region constraints ensuring stable policy updates.
4. Utilization of advanced policy-gradient methods like PPO and GRPO, optimizing policy performance while ensuring efficient and stable training.

Our analytical framework precisely aligns with and generalizes these prevalent RL methods, providing rigorous theoretical foundations and clear guidance for effectively deploying SARL and MARL strategies in sophisticated LLM training scenarios.

## B   DISCUSSION ON THE ANALYTICAL SETTING

### B.1   LLM RL IN PRACTICE

A common paradigm for reinforcement learning (RL) in large language models (LLMs) is reinforcement learning from human feedback (RLHF), where models are fine-tuned using rewards derived from human preference comparisons or proxy reward models. In this paradigm, the LLM functions as an agent that interacts with prompts, generates candidate responses, and receives sequence-level rewards. This aligns with the broader view of RL as optimizing agentic behavior through trial-and-error interaction with an environment.

More recent developments extend beyond single-agent RL (SARL) by leveraging multi-agent RL (MARL). In MARL, the overall task is decomposed into subtasks handled by specialized agents (e.g., retrieval, planning, generation, verification). These agents interact sequentially or in parallel, each contributing to the final outcome while receiving coordinated feedback. Such architectures are increasingly adopted in practice for complex LLM applications because they mirror natural task decomposition and enable specialization and parallelism.

Our analytical setting is designed to capture both paradigms within a unified PAC-learning framework. For SARL, the sequence is generated by a single policy, with rewards assigned at the sequence level, reflecting the RLHF setup. For MARL, we adopt a segment factorization where the output sequence is partitioned into segments, each handled by a specialized agent. This mirrors common multi-agent LLM workflows (e.g., planner–solver–verifier). By formulating both SARL and MARL under the same sequence-level reward function, our setting enables a direct and transparent comparison of their sample complexities.

### B.2 DISCUSSION ON ASSUMPTIONS

The assumptions adopted in our analysis are motivated by standard practices in LLM RL and thus are reasonable approximations of practical training regimes:

- Bounded rewards. In RLHF and related paradigms, rewards are typically normalized or clipped to a bounded interval, commonly $[0, 1]$. This reflects both the probabilistic nature of preference-based rewards and the need for stable optimization.

- Compact parameter space. Practical fine-tuning of LLMs employs regularization strategies such as weight decay, KL penalties around a reference policy, or low-rank adapter updates. These mechanisms constrain the parameter space, justifying our assumption that policy parameters lie in a compact set.

- Lipschitz parameterization. Neural network parameterizations of policies exhibit Lipschitz continuity under standard smoothness assumptions. Additionally, training practices like gradient clipping further enforce bounded sensitivity of action distributions to parameter changes, ensuring the existence of a finite Lipschitz constant.

- Finite horizon. LLM training always imposes a hard cap on sequence length (e.g., maximum token limit), so assuming a finite horizon is consistent with real-world deployments.

Together, these assumptions ensure that the hypothesis class of policies has finite effective complexity, making PAC analysis feasible. Importantly, they reflect routine constraints in contemporary LLM training pipelines rather than idealized simplifications. Consequently, the theoretical results derived under these assumptions can be meaningfully interpreted as guidance for practical RL in LLMs.

## C THE PAC FRAMEWORK AND SAMPLE COMPLEXITY ANALYSIS

The *Probably Approximately Correct* (PAC) learning framework, introduced by Valiant (Valiant, 1984), provides a rigorous foundation for reasoning about the learnability of hypothesis classes under uncertainty. Formally, let $\mathcal{D}$ be an unknown distribution over input space $\mathcal{X}$, and let $\mathcal{H}$ denote a hypothesis class mapping $\mathcal{X}$ to labels or real-valued outputs. Given i.i.d. samples $S = \{x_i\}_{i=1}^n \sim \mathcal{D}^n$, the goal is to identify a hypothesis $h \in \mathcal{H}$ such that its population risk is close to the optimum achievable within $\mathcal{H}$.

**PAC Learnability.** A hypothesis class $\mathcal{H}$ is said to be PAC-learnable if there exists a function $N(\varepsilon, \delta, \mathcal{H})$ such that for all $\varepsilon, \delta \in (0, 1)$, with at least $n \geq N(\varepsilon, \delta, \mathcal{H})$ samples, the empirical risk minimizer $\widehat{h}$ satisfies

$$\Pr\left( L(\widehat{h}) \leq \inf_{h \in \mathcal{H}} L(h) + \varepsilon \right) \geq 1 - \delta,$$

where $L(h) := \mathbb{E}_{x \sim \mathcal{D}}[\ell(h(x))]$ denotes the population risk under a loss $\ell$. Here $\varepsilon$ quantifies the approximation tolerance and $\delta$ the confidence level.

**Sample Complexity.** The function $N(\varepsilon, \delta, \mathcal{H})$ is called the *sample complexity*. In supervised learning, sample complexity is characterized via combinatorial measures such as VC-dimension or via capacity measures such as covering numbers and Rademacher complexity. In reinforcement learning (RL), and in particular the LLM–RL setting, the notion of sample complexity directly corresponds to the number of environment interactions (rollouts or queries) required to guarantee learning of an $\varepsilon$-optimal policy with probability at least $1 - \delta$.

**Application to LLM–RL.** In our analysis, the PAC framework is instantiated for both single-agent RL (SARL) and multi-agent RL (MARL). Policies $\pi \in \Pi$ map text states to action distributions over tokens, and the objective is to maximize the sequence-level reward

$$J(\pi) = \mathbb{E}_{x \sim \mathcal{D}, \, y \sim \pi(\cdot|x)}[R(x, y)].$$

The PAC perspective asks: how many samples are sufficient to guarantee that the empirical maximizer $\widehat{\pi}$ achieves $J(\widehat{\pi}) \geq \sup_{\pi \in \Pi} J(\pi) - \varepsilon$ with probability at least $1 - \delta$? Deriving explicit bounds $N_{\text{SARL}}(\varepsilon, \delta)$ and $N_{\text{MARL}}(\varepsilon, \delta)$ allows us to quantify the comparative sample efficiency of these paradigms.

**Role of Regularity Assumptions.** To obtain finite and non-asymptotic sample complexity bounds, one must control the effective complexity of the policy class $\Pi$. As discussed, in practice, LLM–RL training operates under constraints that naturally satisfy this requirements of bounded rewards, compact parameter space, lipschitz parameterization, and finite horizon. These assumptions imply finite covering numbers for the induced function class $\{f_\theta(x) := \mathbb{E}_{y \sim \pi_\theta}[R(x, y)]\}$, thereby ensuring PAC-learnability. In addition, as we discussed earlier, these assumptions are naturally aligned with the common practice.

**Summary.** The PAC framework thus provides a principled lens for evaluating RL strategies in LLMs, connecting empirical training practices (bounded rewards, regularization, finite horizon) to provable guarantees. By deriving explicit $N(\varepsilon, \delta)$ bounds for SARL and MARL, we identify structural conditions—task decomposition and alignment—that determine when MARL is provably more sample efficient than SARL. This bridges the gap between empirical observations and theoretical understanding, offering a rigorous foundation for the design of agentic LLM systems.

# D PAC SAMPLE COMPLEXITY FOR SARL

In this appendix, we present the proof and derivation for the learning-agnostic PAC sample complexity bound for the single-agent system (SARL) under sequence-level rewards, using covering numbers.

First, let's begin with a brief recall of our SARL setting, which is given as follow: the population objective is

$$J(\boldsymbol{\theta}) = \mathbb{E}_{\boldsymbol{x} \sim \mathbb{D}, \, \boldsymbol{y} \sim \pi_{\boldsymbol{\theta}}(\cdot | \boldsymbol{x})} \big[ R(\boldsymbol{x}, \boldsymbol{y}) \big],$$

Given $n$ samples $\mathcal{S} = \{(\boldsymbol{x}_i, \boldsymbol{y}_i)\}_{i=1}^n$ from $\mathbb{D}$ (e.g., rollouts or logged data),

$$\widehat{J}_n(\boldsymbol{\theta}) = \frac{1}{n} \sum_{i=1}^n R(\boldsymbol{x}_i, \boldsymbol{y}_i), \qquad \widehat{\boldsymbol{\theta}} \in \arg\max_{\boldsymbol{\theta}} \widehat{J}_n(\boldsymbol{\theta}),$$

which is the usual empirical reward maximization (solved in practice by learning algorithms such as gradient descent and stochastic gradient descent). Then under Assumption 1, we can define the value function as follow,

$$f_{\boldsymbol{\theta}}(\mathbf{x}) := \mathbb{E}_{y \sim \pi_{\boldsymbol{\theta}}(\cdot | \mathbf{x})}[R(\mathbf{x}, \mathbf{y})] \in [0, 1], \qquad J(\boldsymbol{\theta}) := \mathbb{E}_{\mathbf{x} \sim \mathbb{D}} \big[ f_{\boldsymbol{\theta}}(\mathbf{x}) \big],$$

with hypothesis (value) class $\mathcal{F} = \{f_{\boldsymbol{\theta}} : \boldsymbol{\theta} \in \Theta\} \subset [0, 1]^{\mathcal{X}}$. Given prompts $\mathbf{x}_1, \dots, \mathbf{x}_n \sim \mathbb{D}$, the estimator is

$$\widehat{J}_n(\boldsymbol{\theta}) := \frac{1}{n} \sum_{i=1}^n f_{\boldsymbol{\theta}}(\mathbf{x}_i).$$

## D.1 FROM PER-STEP CONTROL TO SEQUENCE DISTRIBUTIONS

We begin with deriving some useful results for the sequence level structure. For this, we use the concept of Total Variation (TV) distance between two probability distributions $\mathbb{P}$ and $\mathbb{Q}$ is defined as:

$$TV(\mathbb{P} \mid \mathbb{Q}) = \frac{1}{2} \sum_{x \in \mathcal{X}} |\mathbb{P}(x) - \mathbb{Q}(x)|.$$

**Lemma D.1** (TV control for autoregressive sequence ). *Under Assumption 3 and 4, let $L_{\text{seq}} := T_{\max} L_{\text{step}}$. Then, for any fixed $\mathbf{x}$ and the induced full-sequence laws $\mathbb{P}_{\boldsymbol{\theta}}(\cdot \mid \mathbf{x})$ and $\mathbb{P}_{\boldsymbol{\theta}'}(\cdot \mid \mathbf{x})$, we have*

$$TV(\mathbb{P}_{\boldsymbol{\theta}}(\cdot \mid \mathbf{x}), \, \mathbb{P}_{\boldsymbol{\theta}'}(\cdot \mid \mathbf{x})) \leq L_{\text{seq}} \|\boldsymbol{\theta} - \boldsymbol{\theta}'\|_2.$$

*Proof.* By the definition of TV and Assumption 4, we have that,

$$TV(\mathbb{P}_{\boldsymbol{\theta}}(\cdot \mid \mathbf{x}), \, \mathbb{P}_{\boldsymbol{\theta}'}(\cdot \mid \mathbf{x})) \leq \sum_{t=1}^{T_{\max}} \mathbb{E}\left[ \frac{1}{2} \big\| \boldsymbol{\pi}_{\boldsymbol{\theta}}(\cdot \mid \mathbf{s}_t) - \boldsymbol{\pi}_{\boldsymbol{\theta}'}(\cdot \mid \mathbf{s}_t) \big\|_1 \right]$$

By Assumption 3, we have that

$$\sum_{t=1}^{T_{\max}} \mathbb{E}\left[\frac{1}{2}\big\|\boldsymbol{\pi_\theta}(\cdot \mid \mathbf{s}_t) - \boldsymbol{\pi_{\theta'}}(\cdot \mid \mathbf{s}_t)\big\|_1\right] \leq T_{\max} L_{\text{step}}\|\boldsymbol{\theta} - \boldsymbol{\theta'}\|_2.$$

Subsituting in the definition $L_{\text{seq}} = T_{\max} L_{\text{step}}$, and combining the results above, we get,

$$\text{TV}(\mathbb{P}_{\boldsymbol{\theta}}(\cdot \mid \mathbf{x}), \mathbb{P}_{\boldsymbol{\theta'}}(\cdot \mid \mathbf{x})) \leq L_{\text{seq}}\|\boldsymbol{\theta} - \boldsymbol{\theta'}\|_2$$

$\square$

**Lemma D.2** (Parameter-to-value Lipschitzness)**.** *For any* $\mathbf{x}$,

$$|f_{\boldsymbol{\theta}}(\mathbf{x}) - f_{\boldsymbol{\theta'}}(\mathbf{x})| \leq L_{\text{seq}}\|\boldsymbol{\theta} - \boldsymbol{\theta'}\|_2.$$

*Proof.* The result follow immediately from the definition of $f_{\boldsymbol{\theta}}(\cdot)$ and Lemma. D.1,

$$\begin{aligned}
|f_{\boldsymbol{\theta}}(\mathbf{x}) - f_{\boldsymbol{\theta'}}(\mathbf{x})| &= \left|\mathbb{E}_{\mathbf{y}\sim\mathbb{P}_{\boldsymbol{\theta}}}[\text{R}(\mathbf{x},\mathbf{y})] - \mathbb{E}_{\mathbf{y}\sim\mathbb{P}_{\boldsymbol{\theta'}}}[\text{R}(\mathbf{x},\mathbf{y})]\right|, \\
&\leq \text{TV}(\mathbb{P}_{\boldsymbol{\theta}}, \mathbb{P}_{\boldsymbol{\theta'}}), \\
&\leq L_{\text{seq}}\|\boldsymbol{\theta} - \boldsymbol{\theta'}\|_2
\end{aligned}$$

$\square$

**Lemma D.3** (Parameter-to-function cover)**.** *Let* $\Theta \subset \mathbb{R}^d$ *be a compact parameter set and let* $\{f_\theta : \mathcal{X} \to [0,1]\}_{\theta\in\Theta}$ *be a function class. Assume the parameterization is Lipschitz in the uniform (sup) norm: there exists* $L > 0$ *such that for all* $\theta, \theta' \in \Theta$,

$$\|f_\theta - f_{\theta'}\|_\infty \leq L\,\|\theta - \theta'\|_2.$$

*Then for every* $\varepsilon > 0$,

$$\mathcal{N}(\varepsilon, \{f_\theta : \theta \in \Theta\}, \|\cdot\|_\infty) \leq \mathcal{N}\big(\tfrac{\varepsilon}{L}, \Theta, \|\cdot\|_2\big).$$

*Proof.* Let $\{u_1, \ldots, u_M\}$ be a $(\varepsilon/L)$-cover of $\Theta$ under $\|\cdot\|_2$, i.e., $M = \mathcal{N}(\varepsilon/L, \Theta, \|\cdot\|_2)$ and for every $\theta \in \Theta$ there exists $u_j$ with $\|\theta - u_j\|_2 \leq \varepsilon/L$. By Lemma D.1,

$$\|f_\theta - f_{u_j}\|_\infty \leq L\,\|\theta - u_j\|_2 \leq \varepsilon,$$

so $\{f_{u_1}, \ldots, f_{u_M}\}$ is an $\varepsilon$-cover of the function class in $\|\cdot\|_\infty$. This completes the proof. $\square$

### D.2 SAMPLE COMPLEXITY VIA COVERING THE VALUE CLASS

**Lemma D.4** (Sample Complexity via Covering the Value Class)**.** *Let* $\mathcal{N}(\varepsilon, \mathcal{F}, \|\cdot\|_\infty)$ *denote the covering number of a function class* $\mathcal{F}$ *with respect to the* $\ell_\infty$-*norm, and let* $H_{\mathcal{F}}(\varepsilon) := \log \mathcal{N}(\varepsilon, \mathcal{F}, \|\cdot\|_\infty)$. *Then, under SARL setting and Assumption 2, the following holds:*

$$H_{\mathcal{F}}(\varepsilon) \leq d \log\left(\frac{cL_{\text{seq}}B}{\varepsilon}\right),$$

*where* $c$ *is a universal constant.*

*Proof.* By Lemma D.3, we have the following relationship between the covering numbers of the function class $\mathcal{F}$ and the parameter space $\Theta$:

$$\mathcal{N}(\varepsilon, \mathcal{F}, \|\cdot\|_\infty) \leq \mathcal{N}\left(\frac{\varepsilon}{L_{\text{seq}}}, \Theta, \|\cdot\|_2\right).$$

This bound essentially says that if the parameter space is bounded in the $\ell_2$-norm, the covering number for the value class $\mathcal{F}$ (with respect to the $\ell_\infty$-norm) is at most the covering number for the parameter space $\Theta$ scaled by $L_{\text{seq}}$, which accounts for the sensitivity of the policy to parameter changes.

Next, for an $\ell_2$-ball of radius $B$ in $\mathbb{R}^d$, we can apply standard results in covering number theory. Specifically, we know that:

$$\log \mathcal{N}\left(\eta, \Theta, \|\cdot\|_2\right) \leq d \log \left(\frac{cB}{\eta}\right),$$

where $c$ is a universal constant that depends on the properties of the parameter space.

Combining these results, we obtain the final bound for the covering number of the value class $\mathcal{F}$:

$$H_{\mathcal{F}}(\varepsilon) \leq d \log \left(\frac{cL_{\text{seq}}B}{\varepsilon}\right).$$

This completes the proof. $\qquad\square$

### D.3 Proof of SARL Sample Complexity

Now that we have the main ingredients ready, we present the proof for the main theorem.

*Proof.* Let $\mathcal{C} \subset \mathcal{F}$ be an $\epsilon/4$-net, which means that $\mathcal{C}$ is a finite set of functions such that for any function $f \in \mathcal{F}$, there exists a function $g \in \mathcal{C}$ such that:

$$\|f - g\|_\infty \leq \frac{\epsilon}{4}.$$

In other words, the functions in $\mathcal{C}$ serve as approximations for all the functions in $\mathcal{F}$, with an approximation error of at most $\frac{\epsilon}{4}$.

The size of $\mathcal{C}$, denoted $|\mathcal{C}|$, is given by the covering number $\mathcal{N}(\epsilon/4)$, which counts the minimum number of functions needed to cover $\mathcal{F}$ with an error of $\frac{\epsilon}{4}$.

For any fixed function $g \in \mathcal{C}$, we apply Hoeffding's inequality to the empirical mean $\widehat{\mathbb{E}}_g$ of the function $g$. Hoeffding's inequality provides a bound on the probability that the deviation between the empirical mean and the true mean exceeds a certain threshold. In this case, we are interested in the deviation exceeding $\frac{\epsilon}{2}$, so we use Hoeffding's inequality as follows.

Let $\widehat{\mathbb{E}}_g$ represent the empirical mean:

$$\widehat{\mathbb{E}}_g = \frac{1}{n} \sum_{i=1}^{n} g(x_i),$$

where $x_1, x_2, \ldots, x_n$ are samples drawn from the distribution $\mathbb{D}$. The true expected value of $g$ is:

$$\mathbb{E}_g = \mathbb{E}_{x \sim \mathbb{D}}[g(x)].$$

Hoeffding's inequality tells us that for a bounded random variable $g(x)$ taking values in the interval $[0, 1]$, the probability of the deviation between the empirical mean and the true mean being larger than $\frac{\epsilon}{2}$ is bounded by:

$$\mathbb{P}\left(\left|\widehat{\mathbb{E}}_g - \mathbb{E}_g\right| > \frac{\epsilon}{2}\right) \leq 2 \exp\left(-2n\left(\frac{\epsilon}{2}\right)^2\right) = 2e^{-n\epsilon^2/2}.$$

This means that the probability of the deviation exceeding $\frac{\epsilon}{2}$ decreases exponentially with the number of samples $n$.

Now, we want to bound the probability that the deviation for any function $g \in \mathcal{C}$ exceeds $\frac{\epsilon}{2}$. Since $\mathcal{C}$ is a finite set of functions, we can apply the union bound over all functions in $\mathcal{C}$. The union bound states that the probability of the maximum deviation for all functions $g \in \mathcal{C}$ exceeding $\frac{\epsilon}{2}$ is at most the sum of the probabilities for each function. Specifically:

$$\mathbb{P}\left(\sup_{g \in \mathcal{C}} \left|\widehat{\mathbb{E}}_g - \mathbb{E}_g\right| > \frac{\epsilon}{2}\right) \leq 2\mathcal{N}(\epsilon/4)e^{-n\epsilon^2/2}.$$

Here, $\mathcal{N}(\epsilon/4)$ is the number of functions in the net $\mathcal{C}$, and each function is independently bounded by Hoeffding's inequality.

This means that the probability that the deviation of the empirical mean from the true mean exceeds $\frac{\epsilon}{2}$ for any function in the net $\mathcal{C}$ is bounded by $2\mathcal{N}(\epsilon/4)e^{-n\epsilon^2/2}$.

Now, we move to the deviation for any function $f \in \mathcal{F}$. For any $f \in \mathcal{F}$, we choose a function $g \in \mathcal{C}$ such that the approximation error $\|f - g\|_\infty$ is at most $\frac{\epsilon}{4}$, meaning:

$$\|f - g\|_\infty \leq \frac{\epsilon}{4}.$$

Using the triangle inequality for the difference between the true and empirical means of $f$, we have:

$$\left|\widehat{\mathbb{E}}_f - \mathbb{E}_f\right| \leq \left|\widehat{\mathbb{E}}_f - \widehat{\mathbb{E}}_g\right| + \left|\widehat{\mathbb{E}}_g - \mathbb{E}_g\right| + |\mathbb{E}_g - \mathbb{E}_f|.$$

Now, we bound each term:

• The term $\left|\widehat{\mathbb{E}}_f - \widehat{\mathbb{E}}_g\right|$ is the empirical deviation between $f$ and $g$. Since $\|f - g\|_\infty \leq \frac{\epsilon}{4}$, we have this deviation to be at most $\frac{\epsilon}{4}$.

• The second term $\left|\widehat{\mathbb{E}}_g - \mathbb{E}_g\right|$ is bounded by Hoeffding's inequality, and we know it is at most $\frac{\epsilon}{2}$.

• The third term $|\mathbb{E}_g - \mathbb{E}_f|$ is the true deviation between the expected values of $f$ and $g$, and this term is also bounded by $\frac{\epsilon}{4}$ because $\|f - g\|_\infty \leq \frac{\epsilon}{4}$.

Thus, combining these terms, we get:

$$\left|\widehat{\mathbb{E}}_f - \mathbb{E}_f\right| \leq \frac{\epsilon}{4} + \frac{\epsilon}{2} + \frac{\epsilon}{4} = \epsilon.$$

This shows that, on the complement event (where the deviation does not exceed $\frac{\epsilon}{2}$ for any $g \in \mathcal{C}$), the deviation for any function $f \in \mathcal{F}$ is bounded by $\epsilon$.

Finally, we ensure that with high probability (at least $1 - \delta$), the deviation is bounded by $\epsilon$. To achieve this, we require:

$$2\mathcal{N}(\epsilon/4)e^{-n\epsilon^2/2} \leq \delta.$$

Solving for $n$, we get:

$$n \geq \frac{2}{\epsilon^2}\left(\log \mathcal{N}(\epsilon/4) + \log \frac{2}{\delta}\right),$$

which gives the number of samples required to ensure that, with probability at least $1 - \delta$, the deviation between the true and empirical value functions is bounded by $\epsilon$.

By the result in Lemma D.2, we have,

$$\begin{aligned}
\log \mathcal{N}(\epsilon/4) &= H_{\mathcal{F}}(\varepsilon/4) \\
&\leq d \log\left(\frac{4c\,L_{\text{seq}}B}{\varepsilon}\right) \\
&= d \log\left(\frac{c'\,L_{\text{seq}}B}{\varepsilon}\right).
\end{aligned}$$

Substitute this result in the derivation above, we get,

$$\begin{aligned}
n &\geq \frac{2}{\epsilon^2}\left(\log \mathcal{N}(\epsilon/4) + \log \frac{2}{\delta}\right) \\
&\geq \frac{2}{\epsilon^2}\left(d \log\left(\frac{c'\,L_{\text{seq}}B}{\varepsilon}\right) + \log \frac{2}{\delta}\right)
\end{aligned}$$

This completes the proof.

$\square$

# E   SAMPLE COMPLEXITY FOR MARL

In this appendix, we present the proofs for the sample complexity of MARL. We derive a learning-agnostic (uniform-convergence) PAC bound for a multi-agent system (MARL) under *perfect task decomposition*. Throughout, rewards are bounded in $[0, 1]$ and all expectations are w.r.t. the data distribution $\mathcal{D}$ and the policies' internal randomness.

**Same sequence-level reward, additive form.** To align with SARL, we use the *same* sequence-level reward $\mathrm{R}(\boldsymbol{x}, \boldsymbol{y})$. For readability and to match our PAC analysis, we consider the following two normalized additive form:

$$\overline{\mathrm{R}}_{\mathrm{dep}}(\boldsymbol{x}, \boldsymbol{y}) = \frac{1}{K} \sum_{i=1}^{K} \mathrm{r}_i\big(\boldsymbol{x}, \boldsymbol{y}^{(i)}, \boldsymbol{y}^{(<i)}\big), \tag{E.1}$$

$$\overline{\mathrm{R}}_{\mathrm{indep}}(\boldsymbol{x}, \boldsymbol{y}) = \frac{1}{K} \sum_{i=1}^{K} \mathrm{r}_i\big(\boldsymbol{x}, \boldsymbol{y}^{(i)}\big). \tag{E.2}$$

Eq. 3.1 and Eq. 3.2 capture the spectrum of task decomposition. Eq. 3.1 allows dependence on previous segments (forward coherence) via $\boldsymbol{y}^{(<i)}$. On the other hand, Eq. 3.2 indicate that the reward can be decomposed into independent segments. In later sections, we investigate how the sample complexity of MARL changes under different task decomposition above.

Similar to SARL, the MARL objective is then given by,

$$\mathrm{J}_{\mathrm{MARL}}(\boldsymbol{\pi}_{\boldsymbol{\theta}_{\mathrm{MARL}}}) = \mathbb{E}_{\boldsymbol{x} \sim \mathbb{D}, \, \boldsymbol{y} \sim \boldsymbol{\pi}_{\boldsymbol{\theta}_{\mathrm{MARL}}}(\cdot \mid \boldsymbol{x})}\big[\overline{\mathrm{R}}_{\mathrm{dep/indep}}(\mathbf{x}, \mathbf{y})\big]$$

**Empirical optimization.** Using the same dataset $\mathcal{S} = \{(\boldsymbol{x}_j, \boldsymbol{y}_j)\}_{j=1}^{n}$,

$$\widehat{\mathrm{J}}_{\mathrm{MARL},n}(\boldsymbol{\pi}_{\boldsymbol{\theta}_{\mathrm{MARL}}}) = \frac{1}{n} \sum_{j=1}^{n} \overline{\mathrm{R}}(\boldsymbol{x}_j, \boldsymbol{y}_j), \qquad \widehat{\boldsymbol{\pi}}_{\boldsymbol{\theta}_{\mathrm{MARL}}} \in \arg\max_{\boldsymbol{\pi}} \widehat{\mathrm{J}}_{\mathrm{MARL},n}(\boldsymbol{\pi}_{\boldsymbol{\theta}_{\mathrm{MARL}}}).$$

Any approximate maximizer suffices for the PAC guarantees (statistical bounds separate from optimization error).

**Setup.** There are $K$ agents with policy classes $\Pi_i = \{\pi_{\theta_i} : \theta_i \in \Theta_i\}$. For a prompt $x \sim \mathcal{D}$, the joint policy $\boldsymbol{\pi} = (\pi_{\theta_1}, \ldots, \pi_{\theta_K})$ produces a transcript $Y$ and obtains a bounded reward $R(x, Y) \in [0, 1]$. Let

$$F_{\boldsymbol{\theta}}(x) := \mathbb{E}[R(x, Y)] \in [0, 1], \qquad J(\boldsymbol{\theta}) := \mathbb{E}_{x \sim \mathcal{D}}[F_{\boldsymbol{\theta}}(x)].$$

Given i.i.d. $x_1, \ldots, x_n \sim \mathcal{D}$, define the FI empirical objective

$$\widehat{J}_n(\boldsymbol{\theta}) := \frac{1}{n} \sum_{j=1}^{n} F_{\boldsymbol{\theta}}(x_j).$$

### E.1 PER-AGENT SEQUENCE DISCREPANCY AND VALUE LIPSCHITZNESS

Through similar argument as in the single agent case, we can derive the following similar results for MARL.

**Lemma E.1** (Sequence TV control for agent $i$). *Let $T_{\max}^{(i)}$ denote the maximum sequence length for agent $i$. Under Assumption 3, let $L_{\mathrm{seq}}^{(i)} := T_{\max}^{(i)} L_{\mathrm{step}}$. Then, for any fixed $\mathbf{x}$ and the induced full-sequence laws for agent $i$, $\mathbb{P}_{\boldsymbol{\theta}_i}(\cdot \mid \mathbf{x})$ and $\mathbb{P}_{\boldsymbol{\theta}_i'}(\cdot \mid \mathbf{x})$, we have*

$$\mathrm{TV}\big(\mathbb{P}_{\boldsymbol{\theta}_i}(\cdot \mid \mathbf{x}), \mathbb{P}_{\boldsymbol{\theta}_i'}(\cdot \mid \mathbf{x})\big) \leq L_{\mathrm{seq}}^{(i)} \|\boldsymbol{\theta}_i - \boldsymbol{\theta}_i'\|_2.$$

**Lemma E.2** (Per-agent value Lipschitzness). *For agent $i$ and any $\mathbf{x}$,*

$$|f_{\boldsymbol{\theta}_i}(\mathbf{x}) - f_{\boldsymbol{\theta}_i'}(\mathbf{x})| \leq L_{\mathrm{seq}}^{(i)} \|\boldsymbol{\theta}_i - \boldsymbol{\theta}_i'\|_2.$$

### E.2 PER-AGENT COVERING NUMBERS VIA PARAMETER COVERS

Let $\mathcal{F}_i := \{f_{i,\theta_i} : \theta_i \in \Theta_i\} \subset [0, 1]^{\mathcal{X}}$. Denote covering numbers by $\mathcal{N}(\varepsilon, \cdot, \cdot)$ and $H(\varepsilon) := \log \mathcal{N}(\varepsilon, \cdot, \cdot)$.

**Lemma E.3** (Agent Sample Complexity via Covering the Value Class). *Let $\mathcal{N}(\varepsilon, \mathcal{F}_i, \|\cdot\|_\infty)$ denote the covering number of a function class $\mathcal{F}_i$ with respect to the $\ell_\infty$-norm, and let $H_{\mathcal{F}_i}(\varepsilon) := \log \mathcal{N}(\varepsilon, \mathcal{F}_i, \|\cdot\|_\infty)$. Then, under MARL setting and Assumption 2, the following holds:*

$$H_{\mathcal{F}_i}(\varepsilon) \leq d_i \log\left(\frac{c L_{\mathrm{seq}}^{(i)} B_i}{\varepsilon}\right),$$

*where $c$ is a universal constant.*

**Lemma E.4.** *Let $\Theta \subset \mathbb{R}^d$ be a compact parameter set and let $\{f_\theta : \theta \in \Theta\}$ be a function class such that the parameterization is $L$-Lipschitz in the uniform (sup) norm:*

$$\|f_\theta - f_{\theta'}\|_\infty \leq L\|\theta - \theta'\|_2, \quad \forall \theta, \theta' \in \Theta.$$

*Define the $k$-ary aggregation map*

$$\rho_k(z_1, \ldots, z_k) = \frac{1}{K}\sum_{i=1}^{k} z_i,$$

*which is $L_\rho$-Lipschitz with respect to $\ell_1$, with $L_\rho = 1/K$. Consider the aggregated function class*

$$\mathcal{F}_k := \Big\{(z_1, \ldots, z_k) \mapsto \rho_k(f_{\theta_1}(z_1), \ldots, f_{\theta_k}(z_k)) : \theta_i \in \Theta\Big\}.$$

*Then for every $\varepsilon > 0$,*

$$\mathcal{N}(\varepsilon, \mathcal{F}_k, \|\cdot\|_\infty) \leq \left[\mathcal{N}\left(\frac{K\varepsilon}{L}, \Theta, \|\cdot\|_2\right)\right]^k.$$

*Proof.* By assumption, for each $\theta_i \in \Theta$ there exists $u_i$ from an $(\varepsilon/L)$-cover of $\Theta$ under $\|\cdot\|_2$ such that $\|f_{\theta_i} - f_{u_i}\|_\infty \leq \varepsilon$. Now consider two tuples $\theta_{1:k}$ and $u_{1:k}$. By the Lipschitz property of $\rho_k$ with constant $L_\rho = 1/K$, we have

$$\left\|\rho_k\big(f_{\theta_1}, \ldots, f_{\theta_k}\big) - \rho_k\big(f_{u_1}, \ldots, f_{u_k}\big)\right\|_\infty \leq L_\rho \sum_{i=1}^{k} \|f_{\theta_i} - f_{u_i}\|_\infty.$$

Since each difference is at most $\varepsilon$, this yields

$$\left\|\rho_k\big(f_{\theta_1}, \ldots, f_{\theta_k}\big) - \rho_k\big(f_{u_1}, \ldots, f_{u_k}\big)\right\|_\infty \leq L_\rho \cdot k \cdot \varepsilon = \frac{k}{K}\varepsilon.$$

When $k \leq K$, this is bounded by $\varepsilon$. Thus, the cover of $\Theta$ induces an $\varepsilon$-cover of $\mathcal{F}_k$ in sup norm. Since a cover of $\Theta$ of size $\mathcal{N}(K\varepsilon/L, \Theta, \|\cdot\|_2)$ induces a cover of $\Theta^k$ of size $\left[\mathcal{N}(K\varepsilon/L, \Theta, \|\cdot\|_2)\right]^k$, the result follows. $\square$

**Corollary E.1** (Per-agent specialization used in Theorem 4.2). *For agent $i$, let $\Theta_i \subset \mathbb{R}^{d_i}$ be contained in an $\ell_2$-ball of radius $B_i$, and assume*

$$\|f_{i,\theta_i} - f_{i,\theta_i'}\|_\infty \leq L_{\mathrm{seq},i} \|\theta_i - \theta_i'\|_2 \qquad \forall\, \theta_i, \theta_i' \in \Theta_i.$$

*Then*

$$H(\varepsilon, \mathcal{F}_i, \|\cdot\|_\infty) \leq d_i \log\left(\frac{c\, L_{\mathrm{seq},i} B_i}{\varepsilon}\right),$$

*with the same universal constant $c > 0$ as in Lemma D.3.*

### E.3 PROOF OF MARL SAMPLE COMPLEXITY WITH DEPENDENCE

*Proof.* Fix $\varepsilon, \delta \in (0,1)$ and let $K$ be the number of agents/segments. For $k \in [K]$, write the *partial (normalized) value* after the first $k$ agents as

$$J^{(k)}(\theta_{1:k}) := \mathbb{E}_{x\sim\mathbb{D},\, y\sim\pi_{\theta_{1:k}}(\cdot|x)}\Big[\frac{1}{K}\sum_{i=1}^{k} r_i(x, y^{(i)}, y^{(<i)})\Big],$$

$$\widehat{J}_n^{(k)}(\theta_{1:k}) := \frac{1}{n}\sum_{j=1}^{n}\frac{1}{K}\sum_{i=1}^{k} r_i(x_j, y_j^{(i)}, y_j^{(<i)}).$$

We also define the *stage-$k$ envelope* $G_k(\theta_{1:k-1}) := \sup_{\theta_k} J^{(k)}(\theta_{1:k-1}, \theta_k)$.

Let $\mathcal{F}^{(k)}$ be the function class $\{x \mapsto \frac{1}{K}\sum_{i=1}^{k} r_i(\cdot)\}$ induced by $\{\theta_1, \ldots, \theta_k\}$.

By Lemma E.4 and Lemma E.2, for any $\alpha > 0$, we have,

$$
\begin{aligned}
H\Big(\alpha, \mathcal{F}^{(k)}, \|\cdot\|_\infty\Big) &\leq \sum_{i=1}^{k} d_i \log\left(\frac{c\,L_\rho\,k\,L_{\text{seq},i}B_i}{\alpha}\right) \\
&\leq \sum_{i=1}^{k} d_i \log\left(\frac{c\,L_\rho\,K\,L_{\text{seq},i}B_i}{\alpha}\right).
\end{aligned}
\tag{E.3}
$$

The second inequality just uses $k \leq K$.

Then, for bounded $[0,1]$-valued classes, a standard covering-number generalization bound gives: there exists a universal $C > 0$ such that for any class $\mathcal{F}$ and any $\alpha \in (0,1)$, if

$$
n \;\geq\; C\,\frac{H(\alpha, \mathcal{F}, \|\cdot\|_\infty) + \log(1/\delta')}{\alpha^2},
$$

then with probability at least $1 - \delta'$, $\sup_{f \in \mathcal{F}}\big|\mathbb{E}f - \widehat{\mathbb{E}}_n f\big| \leq \alpha$. Apply this with $\mathcal{F} = \mathcal{F}^{(k)}$, $\alpha = \varepsilon/K$, and $\delta' = \delta/K$. Using equation E.3, it suffices that

$$
n \;\geq\; C\,\frac{\displaystyle\sum_{i=1}^{k} d_i\,\log\!\Big(\frac{c\,L_\rho K L_{\text{seq},i}B_i}{\varepsilon/K}\Big) + \log(K/\delta)}{(\varepsilon/K)^2}.
\tag{E.4}
$$

Because the right-hand side is nondecreasing in $k$, the same $n$ that satisfies equation E.4 for $k = K$ guarantees that, with probability at least $1 - \delta$ (by a union bound over $k = 1, \ldots, K$), the following *simultaneous* uniform deviations hold:

$$
\mathcal{E}_k: \quad \sup_{\theta_{1:k}} \Big| J^{(k)}(\theta_{1:k}) - \widehat{J}_n^{(k)}(\theta_{1:k}) \Big| \;\leq\; \frac{\varepsilon}{K}, \qquad k = 1, \ldots, K.
\tag{E.5}
$$

Define the data-dependent sequence by

$$
\widehat{\theta}_1 \in \arg\max_{\theta_1} \widehat{J}_n^{(1)}(\theta_1), \qquad \widehat{\theta}_k \in \arg\max_{\theta_k} \widehat{J}_n^{(k)}(\widehat{\theta}_{1:k-1}, \theta_k), \quad k = 2, \ldots, K,
$$

and let $\widehat{\theta}_{1:K}$ be the resulting $K$-agent policy.

We prove by induction on $k$ that, on $\mathcal{E}_1 \cap \cdots \cap \mathcal{E}_k$,

$$
G_k(\widehat{\theta}_{1:k-1}) - J^{(k)}(\widehat{\theta}_{1:k}) \;\leq\; \frac{2\varepsilon}{K}.
\tag{E.6}
$$

For $k = 1$, by equation E.5 and the ERM property,

$$
\begin{aligned}
J^{(1)}(\widehat{\theta}_1) &\geq \widehat{J}_n^{(1)}(\widehat{\theta}_1) - \tfrac{\varepsilon}{K} \\
&\geq \sup_{\theta_1} \widehat{J}_n^{(1)}(\theta_1) - \tfrac{\varepsilon}{K} \\
&\geq \sup_{\theta_1} J^{(1)}(\theta_1) - \tfrac{2\varepsilon}{K},
\end{aligned}
$$

which is equation E.6 with $G_1(\emptyset) = \sup_{\theta_1} J^{(1)}(\theta_1)$.

Assume equation E.6 holds up to $k - 1$. For stage $k$, fix $\widehat{\theta}_{1:k-1}$. Because the class $\{\theta_k \mapsto J^{(k)}(\widehat{\theta}_{1:k-1}, \theta_k)\}$ is a subclass of $\mathcal{F}^{(k)}$, equation E.5 implies

$$
\sup_{\theta_k} J^{(k)}(\widehat{\theta}_{1:k-1}, \theta_k) \;\leq\; \sup_{\theta_k} \widehat{J}_n^{(k)}(\widehat{\theta}_{1:k-1}, \theta_k) + \tfrac{\varepsilon}{K} \;=\; \widehat{J}_n^{(k)}(\widehat{\theta}_{1:k}) + \tfrac{\varepsilon}{K} \;\leq\; J^{(k)}(\widehat{\theta}_{1:k}) + \tfrac{2\varepsilon}{K},
$$

which proves equation E.6 at $k$.

Observe that

$$
\sup_{\theta_{1:K}} J^{(K)}(\theta_{1:K}) - J^{(K)}(\widehat{\theta}_{1:K}) \;\leq\; \sum_{k=1}^{K} \big[G_k(\widehat{\theta}_{1:k-1}) - J^{(k)}(\widehat{\theta}_{1:k})\big],
$$

which is obtained by iteratively inserting $\sup_{\theta_k}$ and subtracting/adding $J^{(k)}$. Combining with equation E.6 yields, on $\bigcap_{k=1}^{K} \mathcal{E}_k$,

$$\sup_{\theta_{1:K}} J^{(K)}(\theta_{1:K}) - J^{(K)}(\widehat{\theta}_{1:K}) \leq \sum_{k=1}^{K} \frac{2\varepsilon}{K} = 2\varepsilon.$$

Absorbing the factor 2 into big-$\mathcal{O}$, the learned multi-agent policy is $\varepsilon$-optimal with probability at least $1 - \delta$.

It remains to ensure $n$ satisfies equation E.4 with $k = K$. Using equation E.3 and $\alpha = \varepsilon/K$ gives

$$n = \mathcal{O}\left( \frac{\sum_{i=1}^{K} d_i \log\left(\frac{L_\rho K L_{\mathrm{seq},i} B_i}{\varepsilon}\right) + \log(K/\delta)}{(\varepsilon/K)^2} \right),$$

which is the claimed bound. When $\rho$ is the normalized sum (our Eq. (3.1)), $L_\rho = 1/K$ and the factor $L_\rho K$ inside the logarithm cancels to 1. $\square$

### E.4 PROOF OF SAMPLE COMPLEXITY FOR MARL UNDER INDEPENDENCY

Next, we present the proof for Theorem 4.3. The proof is more straightforward in this case as the agents are independent of each other.

*Proof.* For each agent $i$, define

$$\mathcal{F}_i := \left\{ f_{i,\theta_i}(x) := r_i(x, y^{(i)}; \theta_i) \in [0, 1] \; : \; \theta_i \in \Theta_i \subset \mathbb{R}^{d_i} \right\}.$$

The aggregated (normalized) class is

$$\mathcal{G} := \left\{ g_{\theta_{1:K}}(x) = \frac{1}{K} \sum_{i=1}^{K} f_{i,\theta_i}(x) \; : \; \theta_i \in \Theta_i \; \forall i \right\} \subset [0, 1]^{\mathcal{X}}.$$

Let $J(g) := \mathbb{E}[g(X)]$ and $\widehat{J}_n(g) := \frac{1}{n} \sum_{j=1}^{n} g(X_j)$. If for some $\eta > 0$ we have

$$\sup_{g \in \mathcal{G}} \left| J(g) - \widehat{J}_n(g) \right| \leq \eta, \tag{E.7}$$

then for an ERM $\widehat{g} \in \arg\max_{g \in \mathcal{G}} \widehat{J}_n(g)$ and any maximizer $g^\star \in \arg\max_{g \in \mathcal{G}} J(g)$,

$$\begin{aligned} J(g^\star) - J(\widehat{g}) &\leq [J(g^\star) - \widehat{J}_n(g^\star)] + [\widehat{J}_n(\widehat{g}) - J(\widehat{g})] \\ &\leq 2\eta. \end{aligned}$$

Thus taking $\eta = \varepsilon/2$ gives an $\varepsilon$–optimal policy. It remains to ensure equation E.7 with $\eta = \varepsilon/2$. For any fixed tuple $(\theta_1, \ldots, \theta_K)$,

$$\begin{aligned} \left| J(g_{\theta_{1:K}}) - \widehat{J}_n(g_{\theta_{1:K}}) \right| &= \left| \frac{1}{K} \sum_{i=1}^{K} \left( J(f_{i,\theta_i}) - \widehat{J}_n(f_{i,\theta_i}) \right) \right| \\ &\leq \frac{1}{K} \sum_{i=1}^{K} \sup_{\theta_i \in \Theta_i} \left| J(f_{i,\theta_i}) - \widehat{J}_n(f_{i,\theta_i}) \right|. \end{aligned}$$

Taking the supremum over all $\theta_{1:K}$ yields

$$\sup_{g \in \mathcal{G}} \left| J(g) - \widehat{J}_n(g) \right| \leq \frac{1}{K} \sum_{i=1}^{K} \sup_{\theta_i \in \Theta_i} \left| J(f_{i,\theta_i}) - \widehat{J}_n(f_{i,\theta_i}) \right|. \tag{E.8}$$

Hence it suffices to control each agent's uniform deviation and then average.

By Theorem 4.1 and Lemma E.2, we know that if,

$$n \geq \frac{C}{\varepsilon^2} \left( d_i \log\left(\frac{L_{\text{seq},i} B_i}{\varepsilon}\right) + \log\left(\frac{1}{\delta}\right) \right) \qquad \text{for all } i \in [K], \tag{E.9}$$

then with probability at least $1 - \delta$ (union bound over $i$),

$$\sup_{\theta_i \in \Theta_i} \left| J(f_{i,\theta_i}) - \widehat{J}_n(f_{i,\theta_i}) \right| \leq \varepsilon \qquad \text{simultaneously for all } i \in [K]. \tag{E.10}$$

Combining equation E.8 and equation E.10 gives $\sup_{g \in \mathcal{G}} |J(g) - \widehat{J}_n(g)| \leq \varepsilon$. By Step 1, ERM is $\varepsilon$–optimal with probability $\geq 1 - \delta$, provided $n$ satisfies equation E.9. Since equation E.9 must hold for all $i$, we may replace $d_i$ and $L_{\text{seq},i} B_i$ by their maxima $\widetilde{d} := \max_i d_i$ and $\gamma := \max_i(L_{\text{seq},i} B_i)$ to obtain

$$n \geq \frac{C}{\varepsilon^2} \left( \widetilde{d} \log\left(\frac{\gamma}{\varepsilon}\right) + \log\left(\frac{1}{\delta}\right) \right).$$

This completes the proof. $\qquad \square$

# F    PROOF OF RELATIVE EFFICIENCY

In this appendix, we present the proof and derivation for the relative efficiency results in the paper.

## F.1    PROOF OF PROPOSITION 4.4

*Proof.* From Theorem 4.1 (SARL) and Theorem 4.3 (MARL, independent tasks) we have, suppressing universal constants,

$$N_{\text{SARL}}(\varepsilon, \delta) \asymp \frac{d \log\left(\frac{L_{\text{seq}} B}{\varepsilon}\right) + \log(1/\delta)}{\varepsilon^2}, \tag{F.1}$$

$$L_{\text{seq}} = T_{\max} L_{\text{step}}, \tag{F.2}$$

$$N_{\text{MARL}}(\varepsilon, \delta) \asymp \frac{\widetilde{d} \log\left(\frac{\gamma}{\varepsilon}\right) + \log(1/\delta)}{\varepsilon^2}, \tag{F.3}$$

$$\widetilde{d} = \max_{i \in [K]} d_i, \quad \gamma = \max_{i \in [K]}(L_{\text{seq},i} B_i), \quad L_{\text{seq},i} = T_{\max,i} L_{\text{step}}. \tag{F.4}$$

Under a fair comparison (same backbone/regularization scale), each segment cannot be harder than the monolithic task:

$$\widetilde{d} = \max_i d_i \leq d, \qquad \gamma = \max_i(L_{\text{seq},i} B_i) \leq L_{\text{seq}} B,$$

because $T_{\max,i} \leq T_{\max}$ and each radius $B_i$ is bounded by the SARL radius $B$. Since $x \mapsto \log(x/\varepsilon)$ is increasing on $(\varepsilon, \infty)$, we obtain

$$\widetilde{d} \log\left(\frac{\gamma}{\varepsilon}\right) \leq d \log\left(\frac{L_{\text{seq}} B}{\varepsilon}\right).$$

Plugging this into equation F.4–equation F.2 yields

$$N_{\text{MARL}}(\varepsilon, \delta) \leq \frac{d \log\left(\frac{L_{\text{seq}} B}{\varepsilon}\right) + \log(1/\delta)}{\varepsilon^2}$$
$$\asymp N_{\text{SARL}}(\varepsilon, \delta).$$

i.e.

$$N_{\text{MARL}} \lesssim N_{\text{SARL}}$$

.

Assume $d_i = d/K$ and $T_{\max,i} = T_{\max}/K$ for all $i$. Then $\widetilde{d} = d/K$ and $\gamma = \max_i(L_{\text{seq},i}B_i) = (L_{\text{seq}}/K)\max_i B_i \leq (L_{\text{seq}}B)/K$. Substituting into equation F.4 and dividing by equation F.2 gives

$$\frac{N_{\text{MARL}}}{N_{\text{SARL}}} \leq \frac{\frac{d}{K}\,\log\!\left(\frac{L_{\text{seq}}B}{K\varepsilon}\right) + \log(1/\delta)}{d\,\log\!\left(\frac{L_{\text{seq}}B}{\varepsilon}\right) + \log(1/\delta)}.$$

Using $\log\!\left(\frac{L_{\text{seq}}B}{K\varepsilon}\right) = \log\!\left(\frac{L_{\text{seq}}B}{\varepsilon}\right) - \log K$ and $\log\!\left(\frac{L_{\text{seq}}B}{\varepsilon}\right) - \log K \leq \log\!\left(\frac{L_{\text{seq}}B}{\varepsilon}\right)$, we get the clean upper bound

$$
\begin{aligned}
\frac{N_{\text{MARL}}}{N_{\text{SARL}}} &\leq \frac{\frac{d}{K}\,\log\!\left(\frac{L_{\text{seq}}B}{\varepsilon}\right) + \log(1/\delta)}{d\,\log\!\left(\frac{L_{\text{seq}}B}{\varepsilon}\right) + \log(1/\delta)} \\
&= \frac{1}{K} + \left(1 - \frac{1}{K}\right)\cdot\frac{\log(1/\delta)}{d\,\log\!\left(\frac{L_{\text{seq}}B}{\varepsilon}\right) + \log(1/\delta)}.
\end{aligned}
\tag{F.5}
$$

The second term in equation F.5 is at most $1 - \frac{1}{K}$ and vanishes as soon as the accuracy/complexity term dominates the confidence term, i.e. $d\,\log(L_{\text{seq}}B/\varepsilon) \gg \log(1/\delta)$. Therefore,

$$\frac{N_{\text{MARL}}}{N_{\text{SARL}}} \leq \frac{1}{K} + o\!\left(\frac{1}{K}\right) \qquad \text{(accuracy-dominated regime)},$$

which we summarize as

$$N_{\text{MARL}}/N_{\text{SARL}} \lesssim 1/K.$$

$\square$

### F.2 Proof of Proposition 4.5

*Proof.* By Theorem 4.1 (SARL) and Theorem 4.2 (sequential MARL) there exist universal constants $c_1, c_2 > 0$ such that, for all accuracies $\varepsilon$ in the regime where the logarithms are positive and for all $\delta \in (0,1)$,

$$N_{\text{SARL}}(\varepsilon,\delta) \leq c_1\,\frac{A+L}{\varepsilon^2}, \qquad A := d\,\log\!\left(\frac{L_{\text{seq}}B}{\varepsilon}\right), \quad L := \log\!\left(\frac{1}{\delta}\right), \tag{F.6}$$

$$N_{\text{MARL}}(\varepsilon,\delta) \leq c_2\,\frac{S+L_K}{(\varepsilon/K)^2}, \qquad S := \sum_{i=1}^{K} d_i\,\log\!\left(\frac{L_\rho K\,L_{\text{seq},i}B_i}{\varepsilon}\right), \quad L_K := \log\!\left(\frac{K}{\delta}\right). \tag{F.7}$$

For the normalized additive aggregator (the setting of Theorem 4.2), $\rho(z) = \frac{1}{K}\sum_{i=1}^{K} z_i$ and hence $L_\rho = 1/K$, so $L_\rho K = 1$, and the $K$ inside the logarithm cancels:

$$S = \sum_{i=1}^{K} d_i\,\log\!\left(\frac{L_{\text{seq},i}B_i}{\varepsilon}\right).$$

Divide equation F.7 by equation F.6 and absorb $c_1, c_2$ in $\lesssim$:

$$\frac{N_{\text{MARL}}}{N_{\text{SARL}}} \lesssim K^2 \cdot \frac{S+L_K}{A+L}.$$

Factor the fraction using the identity

$$\frac{x+y}{u+v} = \frac{x}{u}\cdot\frac{1+y/x}{1+v/u},$$

(valid for positive $x, y, u, v$):

$$\frac{S + L_K}{A + L} = \frac{S}{A} \cdot \frac{1 + \dfrac{L_K}{S}}{1 + \dfrac{L}{A}}$$

$$= \underbrace{\frac{\sum_{i=1}^K d_i \log\big(\frac{L_{\text{seq},i} B_i}{\varepsilon}\big)}{d \log\big(\frac{L_{\text{seq}} B}{\varepsilon}\big)}}_{\mathcal{A}(\varepsilon)} \cdot \underbrace{\frac{1 + \dfrac{\log(K/\delta)}{\sum_{i=1}^K d_i \log\big(\frac{L_{\text{seq},i} B_i}{\varepsilon}\big)}}{1 + \dfrac{\log(1/\delta)}{d \log\big(\frac{L_{\text{seq}} B}{\varepsilon}\big)}}}_{\mathcal{C}(\varepsilon, \delta)} \cdot$$

Combining the last two displays gives the desired bound. $\qquad\square$

## G  IMPERFECT TASK ALIGNMENT

In this appendix, we present the proof for the results under imperfect task alignemtn.

### G.1  PROOF OF THEOREM 4.6

*Proof.* Let $J_R(\pi) := \mathbb{E}[R(x, y)]$ and $J_{\overline{R}}(\pi) := \mathbb{E}[\overline{R}(x, y)]$. By the definition of $\alpha$,

$$\forall \pi: \quad \big|J_R(\pi) - J_{\overline{R}}(\pi)\big| \le \alpha, \qquad \Big|\sup_\pi J_R(\pi) - \sup_\pi J_{\overline{R}}(\pi)\Big| \le \alpha. \tag{G.1}$$

Hence for any policy $\widehat{\pi}$,

$$\sup_\pi J_R(\pi) - J_R(\widehat{\pi}) \le \Big[\sup_\pi J_{\overline{R}}(\pi) - J_{\overline{R}}(\widehat{\pi})\Big] + 2\alpha. \tag{G.2}$$

Therefore, to achieve $\varepsilon$–optimality under $R$, it suffices to learn a policy that is $\beta$–optimal under $\overline{R}$ with

$$\beta := \varepsilon - 2\alpha \quad (> 0). \tag{G.3}$$

Write

$$J_{\overline{R}}(\theta_{1:K}) = \frac{1}{K} \sum_{i=1}^K J_i(\theta_i), \qquad \widehat{J}_{\overline{R},n}(\theta_{1:K}) = \frac{1}{K} \sum_{i=1}^K \widehat{J}_{i,n}(\theta_i),$$

where $J_i(\theta_i)$ and $\widehat{J}_{i,n}(\theta_i)$ are the population and empirical expectations of the $i$-th segment reward under agent-$i$ parameters $\theta_i$. By the triangle inequality and the averaging,

$$\sup_{\theta_{1:K}} \big|J_{\overline{R}}(\theta_{1:K}) - \widehat{J}_{\overline{R},n}(\theta_{1:K})\big| \le \frac{1}{K} \sum_{i=1}^K \sup_{\theta_i} \big|J_i(\theta_i) - \widehat{J}_{i,n}(\theta_i)\big|$$

$$\le \max_{i \in [K]} \sup_{\theta_i} \big|J_i(\theta_i) - \widehat{J}_{i,n}(\theta_i)\big|. \tag{G.4}$$

Thus, if for some $\beta > 0$ the right-hand side of equation G.4 is at most $\beta$, then the aggregated uniform deviation is at most $\beta$ as well. Consequently, the sample size required to learn the aggregated objective to accuracy $\beta$ is controlled by the *hardest* agent.

By Theorem 4.3, we immediately get that the aggregated uniform deviation is at most $\beta$ with probability $\ge 1 - \delta$ whenever

$$n \gtrsim \frac{\widetilde{d} \log\big(\frac{K\gamma}{\beta}\big) + \log\big(\frac{1}{\delta}\big)}{\beta^2},$$

$$\widetilde{d} := \max_i d_i,$$

$$\gamma := \max_i (L_{\text{seq},i} B_i).$$

Set $\beta = \varepsilon - 2\alpha$ from equation G.3 and then invoke equation G.2 to convert $\beta$–optimality for $\overline{R}$ into $\varepsilon$–optimality for $R$. This completes the proof. $\qquad\square$

### G.2 PROOF OF PROPOSITION 4.7

*Proof.* By Theorem 4.1 (SARL) and Theorem 4.6 (MARL with alignment), for universal constants $c_1, c_2 > 0$,

$$N_{\text{SARL}}(\varepsilon, \delta) \leq c_1 \frac{\overbrace{d \log\left(\frac{L_{\text{seq}} B}{\varepsilon}\right)}^{=:A} + \overbrace{\log(1/\delta)}^{=:L}}{\varepsilon^2},$$

$$N_{\text{MARL}}(\varepsilon, \delta) \leq c_2 \frac{\overbrace{\widetilde{d} \log\left(\frac{K\gamma}{\varepsilon - 2\alpha}\right)}^{=:C} + \overbrace{\log(1/\delta)}^{=:L}}{(\varepsilon - 2\alpha)^2}.$$

Taking the ratio and absorbing $c_1, c_2$ into $\lesssim$,

$$\frac{N_{\text{MARL}}}{N_{\text{SARL}}} \lesssim \frac{\varepsilon^2}{(\varepsilon - 2\alpha)^2} \cdot \frac{C + L}{A + L}$$

$$= \frac{1}{\left(1 - \frac{2\alpha}{\varepsilon}\right)^2} \cdot \underbrace{\frac{C}{A}}_{= \kappa_d \kappa_\ell} \cdot \underbrace{\frac{1 + \frac{L}{C}}{1 + \frac{L}{A}}}_{= \mathcal{C}_\alpha(\varepsilon, \delta)},$$

which is the expression in the theorem.

If the accuracy terms dominate the confidence terms ($A \gg L$ and $C \gg L$), then $\mathcal{C}_\alpha(\varepsilon, \delta) = 1 + o(1)$, so the ratio condition

$$N_{\text{MARL}} \lesssim N_{\text{SARL}}$$

hold (up to constants) if,

$$\frac{1}{\left(1 - \frac{2\alpha}{\varepsilon}\right)^2} \kappa_d \kappa_\ell \lesssim 1 \iff \kappa_d \kappa_\ell \lesssim \left(1 - \frac{2\alpha}{\varepsilon}\right)^2.$$

Removing the universal constant hidden in $\lesssim$ (by tightening constants in the base theorems) yields the stated "if and only if" criterion in the theorem. $\qquad\square$

## H EXPERIMENT DETAILS

This appendix provides further details on our experimental setup, synthetic task generation, and hyperparameter tuning procedures.

### H.1 TESTBED

Experiments were performed on a Dell PowerEdge C4140 server with the following specifications:

- **CPU:** Dual Intel Xeon Gold 6230 processors (20 cores and 40 threads each).
- **GPU:** Four NVIDIA Tesla V100 SXM2 GPUs, each with 32GB memory and NVLink support.
- **Memory:** 256GB RAM ($8 \times$ 32GB RDIMM modules).
- **Storage:** Dual 1.92TB SSDs (6Gbps SATA interface).
- **Networking:** Dual 1Gbps NICs and a Mellanox ConnectX-5 EX dual-port 40/100GbE QSFP28 adapter with GPUDirect.
- **Operating System:** Ubuntu 18.04 LTS.

### H.2 SYNTHETIC NOISY ARITHMETIC TASKS

We construct synthetic sequence-to-sequence regression tasks based on noisy arithmetic operations. Given a real-valued input sequence $\mathbf{x} = (\mathbf{x}_1, \mathbf{x}_2, \ldots, \mathbf{x}_T)$, the target sequence $\mathbf{y} = (y_1, y_2, \ldots, y_T)$ is generated according to two structural assumptions—independent and dependent subtasks.

### H.2.1  INDEPENDENT NOISY ARITHMETIC TASK

Each output is computed independently based on a local input subsequence. Formally, for each $t$:

$$y_t = \mathbf{w}_t^\top \mathbf{x}_t + \xi_t, \quad \text{with} \quad \xi_t \sim \mathcal{N}(0, \sigma^2).$$

In this scenario, each subtask (predicting $y_t$) is conditionally independent given the input, representing a fully decomposable setting.

### H.2.2  DEPENDENT NOISY ARITHMETIC TASK

Each output depends on both a local input subsequence and previous outputs, introducing dependencies across subtasks:

$$y_t = \mathbf{w}_t^\top \mathbf{x}_t + \text{average}(y^{(<t)}) + \xi_t, \quad \text{with} \quad \xi_t \sim \mathcal{N}(0, \sigma^2).$$

Here, the output at each step incorporates the average of previously generated outputs, creating explicit interdependencies between subtasks.

### H.2.3  COMPARISON OF TASK STRUCTURES

The independent task setting simulates parallelizable subtasks that can be learned separately, while the dependent setting explicitly introduces task coupling, requiring joint reasoning or sequential optimization across subtasks. This difference is key to examining the comparative effectiveness of SARL and MARL strategies in handling different task structures.

## H.3  HYPERPARAMETER TUNING

All models were trained using the Adam optimizer. The primary hyperparameter we tuned was the learning rate $\eta$. We performed a grid search within a predefined range, selecting the optimal value based on performance on an independent validation dataset. This tuning process was conducted separately from the testing phase, ensuring unbiased evaluation of the results.

## H.4  DETAILS OF ADDITIONAL EXPERIMENTS

We have added (1) synthetic alignment sweeps, (2) synthetic agent-count sweeps, and (3) a lightweight real-world LLM experiment on GSM8K. This subsection summarizes the setup of each experiment.

### H.4.1  SYNTHETIC EXPERIMENTS: VARYING THE ALIGNMENT FACTOR $\alpha$

To examine the effect of task alignment, we modify the synthetic task generator to include a tunable dependence parameter $\lambda \in [0, 1]$ that directly controls the induced misalignment factor $\alpha$. For each agent $i$, the subtask output is generated via:

$$y_i = \mathbf{w}_i^\top \mathbf{x}_i + \lambda \cdot \text{average}(y^{(<i)}) + \xi_i, \tag{H.1}$$

where $\xi_i$ is Gaussian noise and $\text{average}(y^{(<i)})$ denotes the mean of all previously generated outputs. The parameter $\lambda$ determines the strength of the dependency:

- $\lambda = 0$ (strong alignment): independent subtasks, yielding $\alpha = 0$;
- $\lambda = 1$ (weak alignment): fully dependent subtasks, yielding large $\alpha$.

We evaluate MARL and SARL across several values of $\lambda$.

### H.4.2  SYNTHETIC EXPERIMENTS: SWEEPING THE NUMBER OF AGENTS $K$

To evaluate the impact of the number of agents, we construct two types of synthetic tasks consistent with the theoretical decompositions:

**Independent subtasks (Eq. (3.2)):**

$$y_i = \mathbf{w}_i^\top \mathbf{x}_i + \xi_i.$$

**Dependent subtasks (Eq. (3.1)):**

$$y_i = \mathbf{w}_i^\top \mathbf{x}_i + \text{average}(y^{(<i)}) + \xi_i.$$

We sweep $K \in \{1, 2, 3, 4\}$ while keeping the *total* model capacity fixed by scaling per-agent dimensionality across runs.

### H.4.3 REAL-WORLD LLM EXPERIMENT ON GSM8K

To test whether our theoretical insights transfer to practical LLM-based RL, we performed a small-scale experiment using the `Qwen` model on the GSM8K math reasoning dataset. We construct two tasks reflecting high- and low-alignment regimes.

**Independent structure (high alignment).** We randomly sample and concatenate two *unrelated* GSM8K math problems into a single input. Agent 1 solves the first problem, and Agent 2 solves the second. The subtasks are independent, inducing a small alignment factor $\alpha$.

**Dependent structure (low alignment).** We decompose each problem into a two-stage pipeline:

$$\text{thinking} \;\rightarrow\; \text{solving}.$$

Agent 1 generates the chain-of-thought, and Agent 2 produces the final answer. Because the solution quality depends heavily on the preceding reasoning, this construction induces strong forward dependence and a large alignment factor.

**Training setups.** For both structures, we compare:

- **SARL**: a single Qwen model trained end-to-end on the complete task.
- **MARL**: two Qwen-based agents, each trained on its own segment of the decomposition.

## I ADDITIONAL DISCUSSION

### I.1 EFFECT OF PARAMETER SHARING ON THE DIMENSION TERMS

A natural question concerns how our analysis treats the parameter dimensions $\{d_i\}_{i=1}^K$ when agents share part of their architectures, such as a common encoder, shared transformer layers, or shared embedding tables. In our framework, the quantity $d_i$ denotes the *effective dimension* of the trainable parameter subset $\Theta_i$ associated with agent $i$. Importantly, $d_i$ does not correspond to the raw number of neural network parameters, but rather to the size of the induced hypothesis class as measured through covering numbers.

**Shared vs. non-shared parameters.** When multiple agents share a common set of parameters (e.g., a shared backbone), the shared block contributes to the covering number of the joint hypothesis class only *once*. Consequently, the effective dimension of a MARL system is not the naive sum $\sum_i d_i$ but instead the covering number of the *union* of all trainable parameter subsets. Our derivations therefore do not assume that agents are parameter-disjoint, nor do they require independent architectures. The covering-number analysis automatically accounts for overlaps in the parameter spaces.

**Impact on sample complexity.** Parameter sharing can only reduce the effective capacity of the MARL hypothesis class. In the independent-subtask regime (Theorem 4.3), the dominant term is $d_e = \max_i d_i$, and shared parameters naturally reduce this quantity. In the dependent-subtask regime (Theorem 4.2), the corresponding term $\sum_i d_i$ decreases whenever agents reuse a shared backbone. In either case, parameter sharing strengthens (rather than weakens) MARL's relative advantage compared to SARL, as it decreases the size of the MARL policy class without altering the SARL complexity.

**Effect on structural conclusions.** The key insight is that our *relative* comparison between SARL and MARL depends on how sample complexity scales with structural factors such as subtask independence, sequential dependence, and task alignment—not on whether parameters are shared or disjoint. Parameter sharing modifies only constant factors through the covering numbers, but it does not alter the qualitative conclusions: MARL retains its benefits under independent decompositions and incurs cumulative costs under dependent ones, regardless of architectural sharing.

**Additional clarifications.** We have added explicit discussion of these considerations in the main text and in the appendix, emphasizing how covering-number methods naturally incorporate shared parameters and why the MARL–SARL comparison remains valid under heterogeneous or partially shared architectures.

### I.2 PRACTICAL ESTIMATION OF THE ALIGNMENT FACTOR $\alpha$

Recall that the alignment factor $\alpha$ (Eq. equation G.1) quantifies the worst-case discrepancy between the original unified reward $R(x, y)$ and the reward induced by the MARL-style independent decomposition:

$$\alpha = \sup_{x,y} \left| R(x, y) - \frac{1}{K} \sum_{i=1}^{K} r_i\left(x, y^{(i)}\right) \right|. \tag{I.1}$$

A small value of $\alpha$ indicates that the decomposition closely matches the underlying task structure, whereas a large $\alpha$ reflects significant misalignment and potentially increased sample complexity (Theorem 4.6 and Proposition 4.7). In this subsection, we describe two practical procedures for estimating $\alpha$ in LLM-based RL settings.

**Monte Carlo estimation.** Given a collection of rollouts $\mathcal{S} = \{(x_j, y_j)\}_{j=1}^{n}$ generated from one or more policies, an empirical upper bound on $\alpha$ can be obtained via

$$\widehat{\alpha}_{\mathrm{MC}} = \max_{(x_j, y_j) \in \mathcal{S}} \left| R(x_j, y_j) - \frac{1}{K} \sum_{i=1}^{K} r_i\left(x_j, y_j^{(i)}\right) \right|. \tag{I.2}$$

In practice, using a high quantile (e.g., the 95th percentile) instead of a strict maximum often yields a more robust and stable estimate, especially in stochastic or high-variance environments. This Monte Carlo approximation can be updated online as additional rollouts are collected.

**Gradient-based maximization.** A complementary approach seeks to approximate the worst-case discrepancy

$$\Delta(x, y) = \left| R(x, y) - \frac{1}{K} \sum_{i=1}^{K} r_i\left(x, y^{(i)}\right) \right|, \tag{I.3}$$

by performing gradient ascent on $(x, y)$ (or on $y$ conditioned on $x$) with respect to $\Delta(x, y)$. This method can reveal adversarial cases or rare task instances where the imposed independent decomposition diverges most sharply from the true reward. Such adversarial examples provide a principled approximation of $\alpha$ beyond what is observed in typical Monte Carlo sampling.

**Discussion.** Both estimation procedures are compatible with standard LLM rollout pipelines and do not require modification to the underlying training framework. While neither approach guarantees an exact evaluation of the supremum defining $\alpha$, together they offer practical and informative estimates that can be used to assess the validity of an imposed task decomposition and to interpret the conditions for MARL–SARL sample-efficiency comparisons.

### I.3 DISCUSSION ON THE USE OF COVERING NUMBERS

A natural question arises from the use of covering-number–based PAC bounds, which are known to be loose in absolute magnitude. We emphasize that in the context of this paper—namely, a *structural comparison* between SARL and MARL—covering numbers remain both appropriate and highly informative.

**Relative efficiency rather than exact sample counts.** Our goal is not to provide numerically tight estimates of the exact number of samples needed for training LLM-based RL systems. Instead, we aim to derive closed-form *scaling laws* that reveal how sample complexity changes as a function of key structural quantities: agent dimensionality $d_i$, number of agents $K$, decomposition structure (independent vs. dependent), and alignment level $\alpha$. For such *comparative* analysis, covering-number bounds are well suited because they directly characterize how the capacity of a policy class scales with its structural components. Tight constants are therefore less important than capturing the correct scaling behavior.

**Advantages over alternative complexity measures.** One might consider alternative complexity notions such as Rademacher complexity. However, Rademacher complexity depends on the *underlying data distribution*, making it difficult to isolate intrinsic structural effects such as reward decomposition or misalignment. In particular, decomposing the reward into $K$ subtasks would interact non-trivially with the rollout-induced distribution, and the misalignment factor $\alpha$ could not be expressed cleanly. In contrast, covering numbers depend on structural properties of the hypothesis class itself, making them substantially more appropriate for analyzing decomposition, sequential dependence, and misalignment—the core drivers of the MARL–SARL comparison.

### I.4 MAPPING THEORETICAL HYPERPARAMETERS TO PRACTICAL LLM–RL QUANTITIES

The theoretical bounds derived depend on a collection of hyperparameters, including the effective dimensions $d_i, d$, the radius parameters $B_i, B$, the Lipschitz constants $L_{\text{step}}, L_{\text{seq}}$, and the sequence length $T$. To facilitate practical interpretation, we provide below a mapping between these quantities and their counterparts in modern LLM–RL pipelines.

**Effective dimensions $d_i$ and $d$.** In our analysis, $d_i$ denotes the *effective trainable dimensionality* of agent $i$'s policy class, not the total number of parameters in a full LLM. In practice, this corresponds to the number of free parameters in the *trainable subset* of the model, such as LoRA modules, adapter layers, or task-specific heads. When multiple agents share parameters (e.g., a common encoder or shared transformer blocks), the shared subset contributes to the covering number *once*. Thus, the effective dimension corresponds to the covering number of the *union* of trainable parameter subsets rather than the naïve sum of raw parameter counts.

**Radius parameters $B_i$ and $B$.** The constants $B_i$ and $B$ capture the radius of the feasible parameter spaces for each agent. In LLM–RL practice, compactness of the parameter domain is induced by standard techniques such as:

- **weight decay** or $\ell_2$ regularization,
- **gradient clipping**, which prevents excessively large updates,
- **KL-regularization** toward a reference model (as used in PPO, GRPO, DPO, and RLHF pipelines),

all of which ensure that the learned policy remains in a bounded neighborhood of the initialization. This matches the compactness assumption required in the PAC analysis.

**Lipschitz constants $L_{\text{step}}$ and $L_{\text{seq}}$.** These constants encode the smoothness of the policy with respect to its parameters at the token level and sequence level, respectively. In practice, these quantities are influenced by:

- **gradient clipping norms**,
- **normalization layers** (LayerNorm, RMSNorm),
- **temperature** and other sampling controls,
- **learning-rate schedules** and optimizer hyperparameters.

Empirically, the Lipschitz constants can be approximated via sensitivity analysis of log-probabilities under small parameter perturbations.

| Symbol | Meaning |
| --- | --- |
| $\mathbf{x} \in \mathcal{X}$ | Input prompt sampled from distribution $\mathbb{D}$. |
| $\mathbf{y} = (a_1, \dots, a_T)$ | Generated output sequence from an LLM policy. |
| $\mathbf{y}^{(i)}$ | Segment $i$ of the decomposed output. |
| $\mathbf{y}^{(<i)}$ | Concatenation of segments 1 to $i-1$ (previous agent outputs). |
| $\pi_\theta$ | SARL policy parameterized by $\theta \in \Theta$. |
| $\pi_{\theta_i}$ | MARL agent-$i$ policy with parameters $\theta_i \in \Theta_i$. |
| $\theta = (\theta_1, \dots, \theta_K)$ | Joint MARL parameter vector. |
| $d$ | Effective dimension of SARL parameter space $\Theta \subset \mathbb{R}^d$. |
| $d_i$ | Effective dimension of agent-$i$ parameter space $\Theta_i \subset \mathbb{R}^{d_i}$. |
| $B, B_i$ | Radius bounds for parameter sets $\|\theta\|_2 \le B$, $\|\theta_i\|_2 \le B_i$. |
| $T_{\max}$ | Maximum generation length in SARL. |
| $T_{\max,i}$ | Maximum generation length for agent $i$ in MARL. |
| $L_{\text{step}}$ | Lipschitz constant of token-level policy w.r.t. parameters. |
| $L_{\text{seq}} = T_{\max} L_{\text{step}}$ | Sequence-level Lipschitz constant for SARL. |
| $L_{\text{seq},i} = T_{\max,i} L_{\text{step}}$ | Sequence-level Lipschitz constant for agent $i$ (MARL). |
| $R(x,y)$ | Unified SARL reward for a full sequence (bounded in $[0,1]$). |
| $r_i(x, y^{(i)}, y^{(<i)})$ | MARL segment reward under *dependent* decomposition (Eq. 3.1). |
| $r_i(x, y^{(i)})$ | MARL segment reward under *independent* decomposition (Eq. 3.2). |
| $R_{\text{dep}}(x,y)$ | Decomposed dependent reward (Eq. 3.1). |
| $R_{\text{indep}}(x,y)$ | Decomposed independent reward (Eq. 3.2). |
| $J(\pi)$ | Population reward objective $\mathbb{E}_{x,y}[R(x,y)]$ (SARL). |
| $J_{\text{MARL}}(\pi)$ | MARL population objective using $R_{\text{dep/indep}}$. |
| $\widehat{J}_n(\pi)$ | Empirical objective from $n$ samples. |
| $N_{\text{SARL}}(\varepsilon, \delta)$ | PAC sample complexity for SARL ). |
| $N_{\text{MARL}}(\varepsilon, \delta)$ | PAC sample complexity for MARL . |
| $K$ | Number of agents (segments) in MARL. |

Table 1: Summary of notation used throughout the paper.

**Sequence length** $T$. The parameter $T$ in our bounds corresponds to the maximum number of decoding steps (or reasoning steps) undertaken within each agent's subtask. In practical LLM–RL settings, this is determined by:

- **context window limits**,
- **maximum generation length** in decoding,
- **structure of decomposed subtasks** (e.g., planner output length, reasoning-chain length, solution depth).

