# OpenReview forum: "When Do Multi-Agent Systems Outperform? Analysing the Learning Efficiency of Agentic Systems"
_ICLR.cc/2026/Conference — Submitted to ICLR 2026_

### Official Review · Reviewer_osy4 · 2025-10-27

**Soundness:** 3
**Presentation:** 3
**Contribution:** 2
**Rating:** 4
**Confidence:** 4

**Summary:**

This paper investigates a fundamental theoretical question: under what conditions does Multi-Agent Reinforcement Learning (MARL) outperform Single-Agent Reinforcement Learning (SARL) in the context of large language model (LLM) training? The authors employ the Probably Approximately Correct (PAC) learning framework to derive sample complexity bounds for both SARL and MARL under different task decomposition scenarios. The main theoretical contributions include: (1) explicit sample complexity characterizations for SARL (Theorem 4.1) and MARL under dependent (Theorem 4.2) and independent (Theorem 4.3) subtask decompositions; (2) formal analysis showing that MARL achieves superior sample efficiency when tasks decompose into independent subtasks, with efficiency gains up to 1/K for K homogeneous agents (Proposition 4.4); and (3) quantification of task misalignment effects through an alignment factor α (Theorem 4.6, Proposition 4.7).

The theoretical analysis is conducted within a well-defined framework with standard regularity assumptions (bounded rewards, compact parameter spaces, Lipschitz parameterization, finite horizon). However, the empirical validation is limited to synthetic noisy arithmetic regression tasks, which raises significant concerns about the practical applicability and real-world relevance of the theoretical insights, particularly given the paper's explicit motivation of LLM agentic systems.

**Strengths:**

### 1. **Addresses an Important and Timely Question**
The paper tackles a fundamental theoretical gap in understanding when and why MARL outperforms SARL in LLM training scenarios. This question is highly relevant given the growing adoption of multi-agent LLM systems in practice (retrieval-augmented generation, tool-augmented generation, multi-step reasoning), yet the lack of theoretical understanding. The motivation is well-articulated and the research question is clearly formulated.

### 2. **Rigorous Theoretical Framework**
The PAC-based analysis is technically sound and provides explicit, non-asymptotic sample complexity bounds. The use of covering numbers and uniform convergence arguments is appropriate and correctly applied. The proofs appear rigorous (based on the appendix), and the assumptions (bounded rewards, compact parameters, Lipschitz continuity, finite horizon) are reasonable and well-justified in Appendix B as aligned with practical LLM training regimes (weight decay, KL penalties, gradient clipping, sequence length limits).

### 3. **Clear Characterization of Task Structure Effects**
The distinction between independent (Equation 3.2) and dependent (Equation 3.1) subtask decompositions is insightful. The theoretical results cleanly show that:
- For independent subtasks, MARL complexity is dominated by the hardest agent: O(d̃ log(γ/ε)/ε²)
- For dependent subtasks, MARL complexity scales with cumulative difficulty: O(Σᵢ dᵢ log(·)/(ε/K)²)
This provides a principled way to reason about task decomposition strategies.

### 4. **Novel Alignment Analysis**
The introduction of the task alignment factor α (Section 4.3) and the characterization of conditions under which MARL remains advantageous despite misalignment (Proposition 4.7) is a valuable theoretical contribution. The condition κ_d κ_ℓ ≤ (1 - 2α/ε)² provides actionable guidance for practitioners.

### 5. **Well-Written and Well-Organized**
The paper is clearly written with good structure. The technical presentation is accessible, with intuitions provided alongside formal statements. The progression from problem setup to main results to alignment analysis is logical.

**Weaknesses:**

This is the paper's most significant weakness and the primary reason for my recommendation to reject. The empirical study (Section 4.4) consists solely of synthetic noisy arithmetic regression tasks:

```
y_i = w_i^T x_i + ξ_i  (independent)
y_i = w_i^T x_i + average(y_(<i)) + ξ_i  (dependent)
```

**Why this is problematic:**

a) **Misalignment with Motivation**: The paper is explicitly motivated by LLM agentic systems, citing applications like retrieval-augmented generation, tool-augmented generation, and multi-step reasoning. Yet there is zero validation on any actual LLM task. The gap between toy linear regression and real LLM training is enormous.

b) **Questionable Relevance**: The assumptions underlying the theory (autoregressive text generation, sequence-level rewards, policy gradient methods like PPO/GRPO mentioned in Appendix A) are completely absent from the experiments. The synthetic tasks do not involve:
   - Language models or text generation
   - Sequential decision-making with discrete actions
   - Reward signals typical of RLHF scenarios
   - Any of the complexity inherent in LLM training

c) **Trivial Task Structure**: The arithmetic tasks are so simplified that they fail to stress-test the theoretical predictions. For instance:
   - The "independent" case is literally independent linear regressions
   - The "dependent" case uses a simple averaging operation
   - There's no exploration, no credit assignment challenges, no compositional complexity

d) **Missing Key Experimental Questions**: The experiments don't address questions critical to the theory's practical utility:
   - How do real LLM tasks align with the independent/dependent decomposition spectrum?
   - Can task decomposition structure be identified in practice?
   - Do the theoretical sample complexity ratios hold in realistic settings?
   - What about partial dependencies or more complex task structures?

**What would be needed**: At minimum, the paper should include experiments on 1-2 real LLM tasks demonstrating:
- A task that naturally decomposes independently (e.g., retrieval + generation)
- Sample efficiency comparisons matching theoretical predictions
- Connection to actual RLHF or policy gradient training

Without this, the paper reads as a theoretical exercise disconnected from its stated application domain.

**Questions:**

1. **Why no real LLM experiments?** This is the elephant in the room. The paper is motivated by LLM agentic systems and cites extensive LLM applications. Why not validate the theory on even a single real LLM task? For example:
   - Code generation with retrieval (MARL: retrieval agent + generation agent vs. SARL: end-to-end)
   - Multi-step reasoning on GSM8K (MARL: decomposed steps vs. SARL: direct)
   - Tool-augmented QA (MARL: tool-use agent + QA agent vs. SARL: integrated)

2. **What is the theory-practice gap?** Given that the bounds are based on covering numbers (which are known to be loose), what is the expected gap between theoretical predictions and practical sample requirements? Can you provide any empirical evidence from the synthetic experiments that the theoretical bounds are not vacuous?

---

> ### Author Response · Authors · 2025-11-24
> **Response to Reviewer osy4 (1/3)**
>
> Dear Reviewer, we sincerely thank you for the detailed, thoughtful, and thorough feedback. We appreciate your positive assessment and recognition of the paper’s theoretical soundness, presentation, and contribution. We respond below to each concern and question and outline the concrete revisions incorporated into the updated manuscript.
>
> ## **Main Weakness and Questions 1: Limitation of Empirical Study**
>
> Thank you for raising this concern and for offering such detailed suggestions. We sincerely appreciate the opportunity to clarify our methodological choices and to improve the empirical component of the paper.
>
> **Motivation for the synthetic empirical study.**
>
> Our main objective in this work is to resolve the *contradicting empirical findings* on when MARL outperforms SARL—a contradiction that persists in the LLM-RL literature from a theoretical perspective. Because our contribution is fundamentally theoretical, our initial submission followed the common practice in theory-oriented papers (e.g., [1–4]) of using synthetic experiments to isolate *statistical phenomena*—such as independence vs. dependence of subtasks and the misalignment factor—without the confounding effects of architecture, optimization instability, reward shaping heuristics, or scale-specific behaviors of LLMs. The synthetic setup allows us to control key variables in a way that is extremely challenging with real LLM systems, where these quantities are tightly entangled with architectural decisions.
>
> In this sense, our synthetic design was intentionally chosen to cleanly reflect the theoretical constructs studied in Sections 4.1–4.3, enabling a transparent validation of the *structural* predictions of our theory. The structure of our synthetic tasks mirrors the key decomposition assumptions made in the theory: **Independent subtasks** correspond exactly to the factorized reward structure of Eq. (3.2). **Dependent subtasks** mirror the forward-dependence structure of Eq. (3.1). This controlled environment allows us to verify the theoretical predictions—such as the gain, the  penalty, and the effect of alignment—that would be difficult to isolate in end-to-end LLM settings.
>
> We hope that you could kindly reconsider the empirical expectations with this theoretical framing in mind. We also note that large-scale LLM-based RL experiments require considerable computational resources, which may not be feasible for all theory-focused researchers.

---

> ### Author Response · Authors · 2025-11-24
> **Response to Reviewer osy4 (2/3)**
>
> ### **Expanded experiments included in the revision**
>
> That said, we fully agree that expanding the empirical study strengthens the paper’s practical relevance. Despite limited computational resources and the challenges of running LLM-based experiments, we have made every effort to curate the necessary resources. In the revised version, we have **substantially expanded** the experimental section to directly address the concerns on the empirical study.
>
> **1. Synthetic experiments varying the alignment factor $\alpha$.**
>
> To validate the theoretical role of alignment, we modified the synthetic model with a **tunable dependence parameter** that controls the misalignment factor $\alpha$. Specifically, for each agent $i$, we define the subtask output as:
>
> $$y_i = \mathbf{w}_i^\top \mathbf{x}_i + \lambda \cdot \mathrm{average}(y^{(<i)}) + \xi_i $$
>
> where $\lambda$ controls the strength of dependence: $\lambda = 0$ yields **perfect alignment** (independent tasks; $\alpha = 0$). $\lambda = 1$ yields **strong misalignment** (high dependence; large $\alpha$). We added an experiment that explicitly varies task alignment by tuning the degree of dependence in the synthetic model. As shown in Figure 1(c):
>
> - When the alignment is **strong** $\lambda=0.1$(small $\alpha$), MARL behaves similarly to SARL.
> - When the alignment is **weak** $\lambda=1$ (large $\alpha$), the performance gap widens, with SARL outperforming MARL.
>
> This validates our theoretical prediction regarding the role of $\alpha$.
>
> **2. Synthetic experiments sweeping the number of agents $K$.**
>
> For both independent and dependent subtasks (Figure 1(a) and (b)), we systematically increase $K$. The results show:
>
> - In the **independent** case, increasing $K$ amplifies MARL’s advantage.
> - In the **dependent** case, increasing $K$ magnifies MARL’s inefficiency.
>
> These trends closely follow the predicted structural behavior based on our sample complexity analysis.
>
> **3. A real-world experiment on GSM8K.**
>
> To connect theory with practice, we added a lightweight LLM evaluation using the **Qwen2.5-1.5B-Instruct** model on the **GSM8K** math reasoning dataset.
>
> We designed two MARL/SARL task formulations:
>
> 1. **Independent structure (high alignment):**
>     - We concatenate **two math problems** into a single input.
>     - *Agent 1* solves the first problem; *Agent 2* solves the second.
>     - The two subtasks are independent.
> 2. **Dependent structure (low alignment):**
>     - We decompose each problem into a two-stage pipeline: **(thinking → solving)**.
>     - *Agent 1* generates the chain-of-thought; *Agent 2* produces the final answer.
>     - The correctness of the final output depends heavily on the earlier reasoning, creating strong cross-stage dependence.
>
> For each structure, we compare:
>
> - **SARL:** a single Qwen model handling the entire task end-to-end,
> - **MARL:** two Qwen-based agents, each trained on its respective subtask.
>
> Results presented in Figures 1(d) and 1(e) match our theoretical predictions:
>
> - **Dependent case:** SARL significantly outperforms MARL.
> - **Independent case:** MARL achieves higher accuracy with fewer samples.
>
> This demonstrates that the theoretical insights transfer meaningfully to real LLM-RL reasoning workflows.

---

> ### Author Response · Authors · 2025-11-24
> **Response to Reviewer osy4 (3/3)**
>
> ## **Q2. The potential the theory–practice gap with covering number bound**
>
> **Reponse**
>
> Thank you for this insightful question.
>
> You are correct that covering-number–based PAC bounds are generally loose in absolute magnitude. However, for our purposes—**a structural comparison between SARL and MARL**—they remain both appropriate and informative for several reasons.
>
> 1. **Our goal is to compare *relative* sample efficiency, not to predict exact sample counts.**
>
> The covering-number framework enables us to derive **closed-form structural relationships** between MARL and SARL (e.g., the gain under independence, the penalty under dependence, and the misalignment penalty). For such comparative analysis, tightness in the bond is less important than correctly capturing how sample complexity scales with agent dimensionality, task decomposition structure, number of agents, and alignment. These qualitative relationships are precisely what covering-number theory is well suited to uncover.  Covering-number bounds offer interpretability and generality, allowing us to express sample complexity in terms of model capacity, Lipschitz smoothness, and segmented reward structure. This interpretability is crucial for analyzing structural relation and misalignment between SARL and MARL, where alternative complexity measures would be far less transparent.
>
> 2. **Why not use alternative complexity measures?**
>
> We have considered other alternative measure such as Rademacher complexity, which can potentially offer tighter sample complexity bound.  Nevertheless, Rademacher complexity depends on the *underlying data distribution* rather than purely on hypothesis-class structure. As a result:
>
> - decomposing the reward into subtasks would interact non-trivially with the induced data distribution,
> - the alignment cost could not be isolated cleanly,
> - and the resulting bounds would depend heavily on rollout-specific dynamics.
>
> In contrast, covering-number bounds depend primarily on **structural properties of the policy class** and are therefore much better suited to analyzing decomposition, sequential dependence, and misalignment—all of which are the core drivers of MARL–SARL differences.
>
> 3. **Empirical evidence that the bounds capture the correct structural trends.**
>
> Although the bounds might not numerically tight, the empirical study show that the **structural predictions** of our theory hold in practice. Specifically: (i) under independent subtasks, MARL exhibits the predicted improvement as $K$ increases; (ii) under dependent subtasks, MARL degrades with $K$; and (iii) when varying the alignment level (increasing $\alpha$), MARL's relative performance to SARL decreases as predicted.
>
> These consistent trend-level agreements indicate that the covering-number bounds are far from vacuous---they correctly capture the key structural behaviors that distinguish SARL and MARL.
>
> **Revision.**
>
> In the revised paper, we have added a dedicated discussion (Appendix I.3) clarifying why covering-number–based bounds are appropriate for our comparative objective.
>
> Thank you again for highlighting this important point; the added discussion improves the clarity and interpretability of our theoretical framework.
>
> ---
>
> # **Closing**
>
> Thank you again for the thoughtful suggestions. We believe the expanded empirical validation and strengthened discussion on the chosen theoretical framework substantially improve the paper’s clarity and practical relevance. We hope these revisions address your concerns and lead to a positive re-evaluation. Please let us know if there are any remaining questions.
>
> [1] Provably efficient RL with Rich Observations via Latent State Decoding, ICML 2019
>
> [2] Scaling Laws in Linear Regression: Compute, Parameters, and Data, NIPS 2024
>
> [3] Reconciling modern machine learning practice and the bias-variance trade-off, PNAS
>
> [4] Is Q-learning Provably Efficient?, NIPS 2018

---

### Official Review · Reviewer_oXhZ · 2025-10-28

**Soundness:** 3
**Presentation:** 4
**Contribution:** 3
**Rating:** 6
**Confidence:** 4

**Summary:**

This paper focuses on comparing the sample efficiency advantages of MARL and SARL approaches in LLM agentic systems. Based on the PAC framework, it formally defines the LLM sequential decision problem and presents a comparison of the efficiency of MARL and SARL under different subtask conditions, along with quantitative analysis and verification. The conclusions are validated using synthetic tasks.

**Strengths:**

LLM agentic systems have recently been frequently explored for complex task processing based on MARL. While many empirical algorithms have emerged for MARL-based approaches, theoretical guidance is currently lacking.

This paper presents a rigorous theoretical derivation and, in the LLM agentic setting, provides a systematic PAC comparison and testable thresholds for SARL vs. MARL. In particular, the introduced "task alignment" factor, $\alpha$, quantifies the sample cost of MARL strategies in complex real-world situations, providing strong theoretical guidance.

**Weaknesses:**

1.	Experimental Verification: First, I think there is a gap between the experimental verification and and motivation & background, which are based on a complex LLM agentic system. Section 4.4 only uses a lightweight synthetic linear task. The "dependence" in the synthetic experiment (i.e., the current output depends on the mean of the previous output) is completely different from the complex semantics, logic, and state dependencies in the LLM agentic system. Therefore, while this experiment mathematically verifies the theory, it does not prove that the theory can be transferred to the LLM problem it claims to solve. Second, the experiment lacks some key theoretical verifications. For example, in Section 4.3, the paper proposes a task misalignment and defines a "misalignment factor" α, deriving new sample bounds. However, the experimental section does not verify α and its PAC bounds at all. Furthermore, the paper itself mentions in its limitations that the experiment only tests the two extreme cases of "complete independence" and "complete dependence," lacking exploration of intermediate scenarios and coverage of "partial dependence."

2.	Limitations of Theoretical Assumptions: Based on strong assumptions, the paper models MARL as segmented turn-based, sequence-level single scalar rewards, and text concatenation. This is significantly different from real-world complex agent interactions, such as parallel collaboration and multi-round debate.

**Questions:**

1.	The theoretical bounds depend on hyperparameters such as $d,d_i,B,B_i,L_{\rm step},T_{\rm max}$. How do these quantities relate to the actual number of parameters in a high-dimensional LLM? Can you provide a theoretical and practical connection?

2.	Can you add supplementary experiments to bridge the gap between the paper's background and experiments, and provide additional experimental support for the relevant theory?

---

> ### Author Response · Authors · 2025-11-24
> **Response to Reviewer oXhZ (1/3)**
>
> ## Rebuttal
>
> Dear Reviewer, we sincerely thank you for the positive assessment of soundness, presentation, and contribution, and for highlighting the value of our analysis.  Bewlow, we address each weakness and question, and list specific revisions we included in the updated manuscript.
>
> ## **W1 and Q2: Simplicity of empirical study and additional empirical support**
>
> **Response**
>
> Thank you for this thoughtful and constructive comment.
>
> Our goal in this paper is to answer the core research question—*when does MARL outperform SARL?*—primarily from a **theoretical perspective**, with the empirical study serving as controlled validation of the key statistical phenomena predicted by our analysis. We hope you could kindly consider this theoretical emphasis when evaluating the empirical component.
>
> That said, we fully agree that expanding the empirical study strengthens the paper’s practical relevance. Despite limited computational resources and the challenges of running LLM-based experiments, we have made every effort to curate the necessary resources. In the revised version, we have **substantially expanded** the experimental section to directly address the concerns on the empirical study.
>
> ### **Expanded experiments included in the revision**
>
> **1. Synthetic experiments varying the alignment factor $\alpha$.**
>
> To validate the theoretical role of alignment, we modified the synthetic model with a **tunable dependence parameter** that controls the misalignment factor $\alpha$. Specifically, for each agent $i$, we define the subtask output as:
>
> $$y_i = \mathbf{w}_i^\top \mathbf{x}_i + \lambda \cdot \mathrm{average}(y^{(<i)}) + \xi_i $$
>
> where $\lambda$ controls the strength of dependence: $\lambda = 0$ yields **perfect alignment** (independent tasks; $\alpha = 0$). $\lambda = 1$ yields **strong misalignment** (high dependence; large $\alpha$). We added an experiment that explicitly varies task alignment by tuning the degree of dependence in the synthetic model. As shown in Figure 1(c):
>
> - When the alignment is **strong** $\lambda=0.1$(small $\alpha$), MARL behaves similarly to SARL.
> - When the alignment is **weak** $\lambda=1$ (large $\alpha$), the performance gap widens, with SARL outperforming MARL.
>
> This validates our theoretical prediction regarding the role of $\alpha$.
>
> **2. Synthetic experiments sweeping the number of agents $K$.**
>
> For both independent and dependent subtasks (Figure 1(a) and (b)), we systematically increase $K$. The results show:
>
> - In the **independent** case, increasing $K$ amplifies MARL’s advantage.
> - In the **dependent** case, increasing $K$ magnifies MARL’s inefficiency.
>
> These trends closely follow the predicted structural behavior based on our sample complexity analysis.
>
> **3. A real-world experiment on GSM8K.**
>
> To connect theory with practice, we added a lightweight LLM evaluation using the **Qwen2.5-1.5B-Instruct** model on the **GSM8K** math reasoning dataset.
>
> We designed two MARL/SARL task formulations:
>
> 1. **Independent structure (high alignment):**
>     - We concatenate **two math problems** into a single input.
>     - *Agent 1* solves the first problem; *Agent 2* solves the second.
>     - The two subtasks are independent.
> 2. **Dependent structure (low alignment):**
>     - We decompose each problem into a two-stage pipeline: **(thinking → solving)**.
>     - *Agent 1* generates the chain-of-thought; *Agent 2* produces the final answer.
>     - The correctness of the final output depends heavily on the earlier reasoning, creating strong cross-stage dependence.
>
> For each structure, we compare:
>
> - **SARL:** a single Qwen model handling the entire task end-to-end,
> - **MARL:** two Qwen-based agents, each trained on its respective subtask.
>
> Results presented in Figures 1(d) and 1(e) match our theoretical predictions:
>
> - **Dependent case:** SARL significantly outperforms MARL.
> - **Independent case:** MARL achieves higher accuracy with fewer samples.
>
> This demonstrates that the theoretical insights transfer meaningfully to real LLM-RL reasoning workflows.
>
> Together, these additional experiments provide more comprehensive empirical support for our theoretical findings.

---

> ### Author Response · Authors · 2025-11-24
> **Response to Reviewer oXhZ (2/3)**
>
> ## **W2: Limitations of modelling assumptions**
>
> **Response**
>
> Thank you for raising this thoughtful point. We appreciate the opportunity to clarify both (1) why our formulation is not as limited as it may initially appear, and (2) why such an abstraction is necessary for a rigorous and meaningful theoretical comparison between SARL and MARL.
>
> **1. Our formulation is not as restrictive as it might seem**
>
> The key structure our analysis relies on is **task dependency**, not turn-taking as an inherent property of the system. The turn-based formulation is an *analytical abstraction* that allows us to represent the dependencies among agent outputs in a mathematically tractable way.
>
> Crucially:
>
> - **Any coordination plan can be mapped to an ordered decomposition**, as long as the outputs of agents can be assigned an ordering.
>
>     For instance, in a parallel coordination plan where agent outputs are independent, one may impose an *arbitrary* ordering of agents. This directly yields our MARL formulation with independent subtasks—without loss of generality.
>
> - **Dependent coordination patterns** (e.g., solver depends on planner; verifier depends on solver) correspond exactly to the forward dependency structure in Eq. (3.1).
>
> Thus, the framework is flexible enough to capture a broad space of coordination mechanisms through the lens of *dependency structure*, which is exactly the property our theoretical results are about.
>
> **2. Why such an abstraction is necessary for theoretical analyticality**
>
> Our goal is to provide a *fair and quantitative* comparison between SARL and MARL under the PAC framework. Achieving this requires a delicate balance:
>
> - **Too much modeling flexibility** (e.g., dynamic role switching) would render a rigorous sample-complexity comparison intractable, as the policies would no longer have stable parameterizations or decompositions.
> - **A simplified but expressive abstraction** enables us to isolate the *statistical drivers* of MARL vs. SARL efficiency—namely, subtask independence, sequential dependence, and alignment.
>
> The turn-based abstraction is therefore not intended to fully model the complex LLM multi-agent dynamic, but to provide a *structured and analyzable abstraction* for understanding core statistical effects that extend beyond specific architectures or coordination protocols.
>
> **Revision**
>
> In the revised paper, we added an additional discussion on formulation to explicitly clarify the scope and generality of our abstraction.
>
> Thank you again for this valuable feedback. It helped us significantly improve the clarity of our modeling assumptions and their relationship to practical multi-agent LLM systems.

---

> > ### Author Response · Authors · 2025-11-24
> > **Response to Reviewer oXhZ (3/3)**
> >
> > ### **Q1. How do the bound hyperparameters map to high‑dimensional LLMs in practice?**
> >
> > **Response**
> >
> > Thank you for this helpful and constructive question. We agree that connecting the theoretical quantities to practical LLM training is important for interpretability. In the revision, we have added a mapping table and a detailed discussion illustrating how each theoretical term corresponds to concrete elements in LLM-based RL.
> >
> > Below is a summary of the mapping that will be incorporated into **Appendix I**:
> >
> > - **$d_i, d$** (Theorems 4.1–4.3): These represent the *effective trainable dimensionality* of the policy class, rather than the full LLM parameter count. In practice, this corresponds to effective dimension of the trainable parameters ( e.g., the number of free parameters in LoRA modules, adapters, or other lightweight tuning mechanisms). When agents share parameters (e.g., a shared encoder or shared transformer layers), the shared components are **counted once**. The effective dimension is thus the covering number of the **union** of trainable subsets rather than the naïve sum of raw parameter counts.
> > - **$B, B_i$** (Assumption 2): These correspond to the radius of the feasible parameter sets. In LLM RL, such compactness is induced by: **weight decay**, **gradient clipping**, and especially **KL-regularization** toward a reference policy (as done in PPO/GRPO and other RLHF pipelines). Together, these mechanisms constrain the policy to remain in a compact neighborhood of the reference model.
> > - **$L_{\text{step}}, L_{\text{seq}}$** (Assumptions 3–4): These correspond to the Lipschitz continuity of the policy with respect to its parameters at the token level and the sequence level. In practice, these are controlled by the design decision or hyper-parameter of the learning algorithm, such as gradient clipping norms, normalization layers, token-level temperature scaling, and overall model smoothness. Empirically, one can approximate these quantities via sensitivity analysis of log-probabilities under parameter perturbations.
> > - **$T$** (sequence length): This maps directly to the maximum generated segment length or number of reasoning steps. In practice, this is set by: LLM context limits, max-token constraints during generation, or natural decomposition boundaries in agentic workflows (e.g., planner output length, solver step length).
> >
> > Together, these mappings allow the theoretical quantities to be estimated or bounded in real LLM-RL settings, enabling meaningful comparison to the PAC bounds.
> >
> > ## Closing
> >
> > Thank you again for the encouraging evaluation and thoughtful suggestions. We believe the expanded empirical validation and strengthened modeling discussion substantially improve the paper’s clarity and practical relevance. We hope these revisions address your concerns. Please let us know if there are any remaining questions.

---

### Official Review · Reviewer_geeW · 2025-10-29

**Soundness:** 3
**Presentation:** 2
**Contribution:** 2
**Rating:** 4
**Confidence:** 3

**Summary:**

The paper provides a theoretical analysis of when multi-agent systems outperform single-agent ones in learning efficiency. Using the PAC framework, it derives sample complexity bounds for both settings and shows that multi-agent systems gain advantages when tasks can be decomposed into independent subtasks but lose efficiency when interdependencies increase. A small synthetic experiment supports the theory, illustrating how task alignment determines when multi-agent learning is beneficial.

**Strengths:**

- The paper addresses a valuable and timely problem: providing a theoretical understanding of when multi-agent systems outperform single-agent systems. Such an analysis is needed to ground current LLM-based multi-agent research in solid theory.

- The theoretical foundation is solid. The authors derive PAC-based sample complexity bounds for both MARL and SARL, offering a rigorous comparison of their learning efficiencies.

- The inclusion of a small empirical study adds some empirical support to the theoretical conclusions and helps illustrate the conditions under which multi-agent setups are beneficial.

**Weaknesses:**

- While the theoretical analysis is sound, the setting is overly simplified. The “multi-agent” formulation effectively reduces to a fixed workflow where each agent corresponds to a submodel handling a static subtask. This abstraction misses the richer dynamics of real LLM-based multi-agent systems, where benefits and failures often stem from high-level task decomposition and coordination rather than low-level execution efficiency.

- The conclusions that multi-agent systems help when subtasks are independent and hurt when they are interdependent, though well-derived, are quite intuitive. It offers limited new insight beyond established intuition.

- The empirical study is minimal, involving only a single toy task without details on model training or architecture. Also, despite frequent references to “LLMs,” the experimental setup and simplifications in the theoretical deduction make the work largely disconnected from realistic LLM-based multi-agent scenarios.

**Questions:**

Sea above

---

> ### Author Response · Authors · 2025-11-24
> **Response to Reviewer geeW (1/2)**
>
> Dear Reviewer, we sincrely thank you for the thoughtful assessment, the recognition of the problem’s importance, and the positive evaluation of the theoretical foundation. We sincerely appreciate your constructive feedback and address each weakness below, along with the revisions made in the updated manuscript.
>
> ## **W1. On “overly simplified multi‑agent” modeling**
>
> **Response**
>
> Thank you for raising this thoughtful point. We appreciate the opportunity to clarify both (1) why our formulation is not as limited as it may initially appear, and (2) why such an abstraction is necessary for a rigorous and meaningful theoretical comparison between SARL and MARL.
>
> **1. Our formulation is not as restrictive as it might seem**
>
> The key structure our analysis relies on is **task dependency**, not turn-taking as an inherent property of the system. The turn-based formulation is an *analytical abstraction* that allows us to represent the dependencies among agent outputs in a mathematically tractable way.
>
> Crucially:
>
> - **Any coordination plan can be mapped to an ordered decomposition**, as long as the outputs of agents can be assigned an ordering.
>
>     For instance, in a parallel coordination plan where agent outputs are independent, one may impose an *arbitrary* ordering of agents. This directly yields our MARL formulation with independent subtasks—without loss of generality.
>
> - **Dependent coordination patterns** (e.g., solver depends on planner; verifier depends on solver) correspond exactly to the forward dependency structure in Eq. (3.1).
>
> Thus, the framework is flexible enough to capture a broad space of coordination mechanisms through the lens of *dependency structure*, which is exactly the property our theoretical results are about.
>
> **2. Why such an abstraction is necessary for theoretical analyticality**
>
> Our goal is to provide a *fair and quantitative* comparison between SARL and MARL under the PAC framework. Achieving this requires a delicate balance:
>
> - **Too much modeling flexibility** (e.g., dynamic role switching) would render a rigorous sample-complexity comparison intractable, as the policies would no longer have stable parameterizations or decompositions.
> - **A simplified but expressive abstraction** enables us to isolate the *statistical drivers* of MARL vs. SARL efficiency—namely, subtask independence, sequential dependence, and alignment.
>
> The turn-based abstraction is therefore not intended to fully model the complex LLM multi-agent dynamic, but to provide a *structured and analyzable abstraction* for understanding core statistical effects that extend beyond specific architectures or coordination protocols.
>
> **Revision**
>
> In the revised paper, we added an additional discussion on formulation to explicitly clarify the scope and generality of our abstraction.
>
> Thank you again for this valuable feedback. It helped us significantly improve the clarity of our modeling assumptions and their relationship to practical multi-agent LLM systems.
>
> ## **W2. On “results feel intuitive”**
>
> **Response**
>
> Thank you for this thoughtful comment.
>
> While the conclusion may appear intuitive in hindsight, we emphasize that it has **not been intuitive in practice**. This is evidenced by **contradictory empirical findings** in the literature: several works report that MARL consistently outperforms SARL, while others assert the opposite (see Existing Gap paragraph in the introduction). These inconsistencies highlight that the field lacks a coherent understanding of when MARL is beneficial and when it is not. Clarifying this confusion was one of the primary motivations for our work.
>
> Beyond the high-level statement, our analysis makes the intuition more precise, actionable, and quantitative. For example, our sample-complexity analysis identifies concrete structural quantities—such as per-agent dimensions, sequence lengths, and the decomposition structure—that  determine the relative efficiency of SARL vs. MARL. In addition, we introduce the alignment factor $\alpha$, which quantifies the cost incurred when approximating a dependent task with an independent decomposition. This gives a principled way to evaluate a commonly used but previously poorly understood design choice in LLM-based agentic systems.
>
> Together, these results not only help resolve a long-standing ambiguity in the community but also elevate a seemingly intuitive idea into a rigorous and practically useful framework. The theory not only explains why MARL helps or hurts, but also provides clear guidance on how to choose and design real LLM-RL systems.

---

> ### Author Response · Authors · 2025-11-24
> **Response to Reviewer geeW (2/2)**
>
> ## **W3. On the limitation of empirical study**
>
> **Response & changes.**
>
> Thank you for this thoughtful and constructive comment.
>
> Our goal in this paper is to answer the core research question—*when does MARL outperform SARL?*—primarily from a **theoretical perspective**, with the empirical study serving as controlled validation of the key statistical phenomena predicted by our analysis. We hope you could kindly consider this theoretical emphasis when evaluating the empirical component.
>
> That said, we fully agree that expanding the empirical study strengthens the paper’s practical relevance. Despite limited computational resources and the challenges of running LLM-based experiments, we have made every effort to curate the necessary resources. In the revised version, we have **substantially expanded** the experimental section to address the concerns on the empirical study.
>
> ### **Expanded experiments included in the revision**
>
> **1. Synthetic experiments varying the alignment factor $\alpha$.**
>
> To validate the theoretical role of alignment, we modified the synthetic model with a **tunable dependence parameter** that controls the misalignment factor $\alpha$. Specifically, for each agent $i$, we define the subtask output as:
>
> $$y_i = \mathbf{w}_i^\top \mathbf{x}_i + \lambda \cdot \mathrm{average}(y^{(<i)}) + \xi_i $$
>
> where $\lambda$ controls the strength of dependence: $\lambda = 0$ yields **perfect alignment** (independent tasks; $\alpha = 0$). $\lambda = 1$ yields **strong misalignment** (high dependence; large $\alpha$). We added an experiment that explicitly varies task alignment by tuning the degree of dependence in the synthetic model. As shown in Figure 1(c):
>
> - When the alignment is **strong** $\lambda=0.1$(small $\alpha$), MARL behaves similarly to SARL.
> - When the alignment is **weak** $\lambda=1$ (large $\alpha$), the performance gap widens, with SARL outperforming MARL.
>
> This validates our theoretical prediction regarding the role of $\alpha$.
>
> **2.Synthetic experiments sweeping the number of agents $K$.**
>
> For both independent and dependent subtasks (Figure 1(a) and (b)), we systematically increase $K$. The results show:
>
> - In the **independent** case, increasing $K$ amplifies MARL’s advantage.
> - In the **dependent** case, increasing $K$ magnifies MARL’s inefficiency.
>
> These trends closely follow the predicted structural behavior based on our sample complexity analysis.
>
> **3. A real-world experiment on GSM8K.**
>
> To connect theory with practice, we added a lightweight LLM evaluation using the **Qwen2.5-1.5B-Instruct** model on the **GSM8K** math reasoning dataset.
>
> We designed two MARL/SARL task formulations:
>
> 1. **Independent structure (high alignment):**
>     - We concatenate **two math problems** into a single input.
>     - *Agent 1* solves the first problem; *Agent 2* solves the second.
>     - The two subtasks are independent.
> 2. **Dependent structure (low alignment):**
>     - We decompose each problem into a two-stage pipeline: **(thinking → solving)**.
>     - *Agent 1* generates the chain-of-thought; *Agent 2* produces the final answer.
>     - The correctness of the final output depends heavily on the earlier reasoning, creating strong cross-stage dependence.
>
> For each structure, we compare:
>
> - **SARL:** a single Qwen model handling the entire task end-to-end,
> - **MARL:** two Qwen-based agents, each trained on its respective subtask.
>
> Results presented in Figures 1(d) and 1(e) match our theoretical predictions:
>
> - **Dependent case:** SARL significantly outperforms MARL.
> - **Independent case:** MARL achieves higher accuracy with fewer samples.
>
> This demonstrates that the theoretical insights transfer meaningfully to real LLM-RL reasoning workflows.
>
> Together, these additional experiments provide more comprehensive empirical support for our theoretical findings.
>
> ## **Closing**
>
> Thank you again for the constructive and thoughtful feedback. We believe the clarifications, improved exposition, and expanded empirical results have significantly strengthened the paper. We hope the revisions satisfactorily address your concerns and lead to a positive re-evaluation. Please let us know if there are any further questions.

---

### Official Review · Reviewer_aE3u · 2025-10-30

**Soundness:** 2
**Presentation:** 2
**Contribution:** 2
**Rating:** 4
**Confidence:** 2

**Summary:**

This paper presents a clear theoretical framework for comparing the sample efficiency of MARL and SARL under different task decomposition structures. The topic is important and timely in the context of LLM-based RL. The theoretical analysis is mathematically sound and extends to non-ideal task alignment scenarios, offering practical insights. However, several theoretical assumptions are not clearly stated or sufficiently justified, including the independence among agents, the definition of some symbols, the derivation of certain theorems, and the reliance on specific assumptions. In addition, the empirical study is relatively simplified and does not fully examine the effects of key parameters.

**Strengths:**

The paper addresses an important and timely problem in the context of LLM-based RL by clarifying the theoretical boundary between MARL and SARL applicability. It provides valuable insights into how task decomposition structure influences sample efficiency and extends the discussion to imperfect alignment cases, offering practical relevance beyond ideal theoretical settings.

**Weaknesses:**

1. The paper does not specify whether agents share the same input or have partial observations. Would this assumption affect the validity of the theoretical derivations?
2. Several symbols, such as $r_i$ in Eq.~(3.1), are introduced without clear definitions. Please provide precise definitions to ensure the rigor and clarity of the theoretical reasoning.
3. The paper does not clearly explain why the $K^2$ term in Theorem~4.2 represents a tight bound. The dependence among rewards $r_i$ is not formally defined, so the derivation of this scaling factor lacks a clear justification.
4. Does the analysis assume i.i.d. samples under a fixed distribution? If so, it would be helpful to mention this in the limitations section, since real RL/MARL trajectories are often temporally correlated, which may affect the practical validity of the theoretical bounds.
5. The analysis assumes that agent dimensions $d_i$ can be directly summed. Does this imply independent and non-overlapping parameters? In practice, shared or heterogeneous architectures may violate this assumption and lead to an overestimation of model complexity.
6. The alignment factor $\alpha$ is defined in Eq.~(4.1) but not specified or computed. More explanation is needed on how $\alpha$ could be interpreted or estimated in LLM-based RL settings.
7. The empirical study is too simplified and does not capture the effects of key parameters such as $K$ and $d$, or other factors identified as influential in the theoretical analysis.

**Questions:**

1. Can the above weaknesses be addressed?
2. Releasing the code or benchmark used in the empirical study would improve the paper’s transparency and reproducibility.

---

> ### Author Response · Authors · 2025-11-23
> **Response to Reviewer aE3u (1/4)**
>
> Dear Reviewer, we sincerely thank you for the insightful questions, and thoughtful and constructive feedback. We are very gald that you find the problem important and the analysis timely and practically relevant. Below, we address each concern in detail and will incorporate all clarifications and improvements into the revised manuscript.
>
> ## **W1. Shared input vs. partial observations**
>
> **Response:**
>
> Thank you for raising this question.
>
> In our theoretical setup, all agents are assumed to condition on the same global prompt $\mathbf{x}$ (i.e., together solving a ``global task'') and on the conversation history available up to their turn. This matches common multi-agent LLM pipelines (e.g., planner → solver → verifier), where later agents observe earlier outputs. Thus, agents do **not** have identical observations, but rather **monotonically increasing** context as the interaction progresses.
>
> **Regarding the theoretical results**
>
> Our PAC-style derivations rely only on the Lipschitz continuity and covering-number properties of each agent’s policy class. These bounds are agnostic to the exact form of the observation space. If one wishes to model a different observation structure—e.g., agent-specific partial observations $o_i$—the proofs remain largely unchanged after redefining the per-agent value function as $f_{i,\theta_i}(o_i)$. Only the associated Lipschitz constants and covering numbers would shift accordingly; the overall inequalities and scaling laws still hold.
>
> **Revision:**
>
> We will add clarifications in Sec. 3.2 and include a remark emphasizing that partial observability affects only constant factors, not the validity or form of the sample-complexity bounds.
>
> ## **W2. Missing symbol description in Eq. (3.1)**
>
> **Response:**
>
> Thank you for highlighting this important point.
>
> During our paper revision, some explicit explanations distinguishing the global reward $R$ and the per-agent subtask rewards $r_i$ were unintentionally removed. We apologize for the confusion. To clarify:
>
> - $R(x, y)$ denotes the **global sequence-level reward** for the entire task.
> - $r_i(x, y^{(i)}, y^{(<i)})$ denotes the **local reward** associated with the i-th subtask/agent, explicitly showing its dependence on the $i$-th agent output and all preceding agents.
>
> These definitions underlie Eq. (3.1), where the global reward is decomposed into subtask-level contributions.
>
> **Revision:**
>
> - We have restore the full explanation of Eq. (3.1), explicitly defining all symbols used.
> - We carefully review the entire manuscript to ensure every notation is introduced clearly at first use.
> - We additionally include a **notation table** in the appendix to help readers follow the formalism more easily.
>
> Thank you again for pointing out this omission—this clarification will improve the paper’s rigor and readability.
>
> ## **W3. Tightness of the $K^2$ factor in Theorem 4.2**
>
> **Response:**
>
> Thank you for this insightful questions!
>
> In the following, we clarify the dependency among rewards and the origin of the $K^2$ factor
>
> - **Dependence structure**
>
> The dependence structure we analyze is the one captured in Eq. (3.1), where the reward for the $i$-th agent satisfies $r_i(x, y^{(i)}, y^{(<i)})$ meaning each agent depends on the outputs of previous agents. Our formulation is intentionally general and does not assume any specific functional form for this dependence, allowing the results to apply to a broad class of sequential multi-agent settings. We believe that incorporating more fine-grained dependence models could yield sharper results, and we identify this as an interesting direction for future work.
>
> - **Origin of $K^2$**
>
> The $K^2$ term arises from two interacting effects in the dependent case:
>
> **Error propagation.**
>
> Since agent can depends on other agents, approximation error introduced early in the sequence can propagate downstream. To ensure the overall deviation remains below $\varepsilon$, we must bound per-agent error at order $\varepsilon/K$.
>
> **Uniform convergence scaling.**
>
> In PAC analysis, uniform convergence bounds scale with the *inverse square* of the accuracy parameter. Replacing $\varepsilon$ with $\varepsilon/K$ therefore introduces a factor of $K^2$
>
> Thus, the $K^2$ dependence reflects the intrinsic statistical cost of sequentially dependent subtasks rather than an artifact of our proof technique.
>
> **Revision**
>
> - We have expanded the explanation of the dependence structure in Eq. (3.1).
> - We have added a discussion after Theorem 4.2 clarifying the precise origin of the $K^2$ scaling.
>
> Thank you again for the helpful suggestion—this clarification strengthens the theoretical transparency of our results.

---

> ### Author Response · Authors · 2025-11-23
> **Response to Reviewer aE3u (2/4)**
>
> ## **W4. i.i.d. samples vs. correlated RL trajectories**
>
> **Response:**
>
> Thank you for raising this interesting point.
>
> Our analysis does assumes that the training samples $(x, y)$ are drawn i.i.d. from a fixed distribution $D$. This matches the dominant training regime in LLM-based RL settings such as RLHF and offline RLHF-style rollouts, where prompts are sampled from a static dataset or replay buffer and sequence-level rewards are evaluated independently.
>
> We agree that in fully interactive (online) RL environments, trajectories may be temporally correlated. In such cases, additional assumptions (e.g., mixing-time conditions or Martingale concentration inequalities) would be required, and the constants in the bounds would depend on the underlying mixing rate. Extending the analysis to such a setting renders as an interesting future study.
>
> **Revision**
>
> - We have added additional paragraph in Section 5 (Concluding Discussion) explicitly stating the i.i.d. sampling assumption and discussing how the analysis could be extended to correlated trajectories using standard mixing-based PAC techniques.
>
> ## **W5. Summation of per-agent dimensions**
>
> **Response:**
>
> Thank you for raising this important point.
>
> **What $d_i$ represents.**
>
> In our analysis, $d_i$ denotes the *effective dimension* (e.g., the tuneable rank in LoRA) of the parameter space $\Theta_i$ associated with agent $i$. It is not tied to a specific architecture or parameterization style; rather, it reflects the size of the hypothesis class induced by the trainable parameters of each agent.
>
> **Whether this leads to overestimation.**
>
> When agents share parameters—e.g., via a common encoder, shared transformer layers, or shared embeddings—the shared component is **counted only once** in the covering number of the joint hypothesis class. In such cases, the effective dimension of the overall MARL system is not the naive sum $\sum_i d_i$, but the covering number of the union of trainable parameter sets. Thus, our derivation does not intrinsically assume non-overlapping parameters, nor does it require each agent to be fully independent.
>
> **Why this does not affect our main comparison.**
>
> Our result compare MARL and SARL in terms of **relative** sample complexity. Even if parameter sharing reduces the effective MARL complexity, this only strengthens MARL’s advantage because it further decreases the dominating term $ d_e = \max d_i $ in the independent case or $\sum_i d_i$ in the dependent case. The qualitative conclusions—e.g., MARL benefits under independent decompositions and incurs a cumulative cost under dependent ones—do not rely on whether agents share components. Only the constants change, not the structural comparison.
>
> **Revision**
>
> - We have added a clarification in Section 3 to explain the dimension factor $d$.
> - We have added a dicussion in Appendix I on how parameter sharing is handled through covering numbers, and why shared parameters do not affect the theoretical comparison between SARL and MARL.
>
> Thank you again for this helpful point—this clarification strengthens the rigor and generality of our theoretical setup.

---

> ### Author Response · Authors · 2025-11-24
> **Response to Reviewer aE3u (3/4)**
>
> ## **W6. Interpretation and estimation of the alignment factor $\alpha$**
>
> **Response:**
>
> Thank you for this helpful comment.
>
> We agree that $\alpha$ deserves a clearer explanation and more guidance on practical estimation.
>
> **Interpretation of $\alpha$.**
>
> As defined in Eq. (4.1), the alignment factor $\alpha$ measures the **worst-case discrepancy** between the original unified reward and the reward obtained by forcing an independent subtask decomposition. Intuitively, $\alpha$ quantifies *how well the MARL decomposition aligns with the true underlying task structure*. A small $\alpha$ means the decomposition is faithful; a large $\alpha$ indicates that enforcing independence may distort the reward and degrade sample efficiency, as reflected in Theorem 4.6 and Proposition 4.7
>
> **Practical estimation of $\alpha$ in LLM-based RL.**
>
> In practice, $\alpha$ can be estimated using either of the following methods:
>
> - **Monte Carlo estimation:** Sample rollouts from the current policy (or multiple policies) and compute $\widehat{\alpha} = \max_{(x,y) \in \mathcal{S}} \left| R(x,y) - \frac{1}{K} \right|.$ Using a high quantile (e.g., 95th percentile) often provides a robust upper bound.
>
> - **Gradient-based maximization:** Define $\Delta(x,y) = \left| R(x,y) - \frac{1}{K} \sum_i r_i(x, y^{(i)}) \right|$ as an objective and use gradient ascent over $x,y$ to approximate the maximizer. This may reveal worst-case scenarios in complex LLM pipelines.
>
> Both procedures are compatible with standard LLM rollouts and allow empirical estimation of the factor $\alpha$.
>
> **Revision:**
>
> - We have expanded the explanation of $\alpha$ following Eq. (4.1) to clarify its meaning and role in the theory.
> - We have added a short subsection in the appendix describing practical methods—Monte Carlo estimation and gradient-based maximization—for estimating $\alpha$ in real LLM-agent setups.

---

> ### Author Response · Authors · 2025-11-24
> **Response to Reviewer aE3u (4/4)**
>
> ## **W7. Simplified empirical study**
>
> **Response:**
>
> Thank you for the helpful suggestion.
>
> Our original intent was to address the research question—*when does MARL outperform SARL?*—primarily from a **theoretical perspective**, with experiments serving as supporting validation rather than full-scale empirical benchmarking. We hope you could kindly keep this theoretical emphasis in mind when evaluating the experimental component.
>
> That said, we fully agree that expanding the empirical study to vary key structural parameters would strengthen the practical relevance of our results. To address this, we have substantially expanded the empirical section in the revised paper.
>
> ### **Expanded experiments included in the revision**
>
> **1. Synthetic experiments varying the alignment factor $\alpha$.**
>
> To validate the theoretical role of alignment, we modified the synthetic model with a **tunable dependence parameter** that controls the misalignment factor $\alpha$. Specifically, for each agent $i$, we define the subtask output as:
>
> $$y_i = \mathbf{w}_i^\top \mathbf{x}_i + \lambda \cdot \mathrm{average}(y^{(<i)}) + \xi_i $$
>
> where $\lambda$ controls the strength of dependence: $\lambda = 0$ yields **perfect alignment** (independent tasks; $\alpha = 0$). $\lambda = 1$ yields **weak malignment** (high dependence; large $\alpha$). We added an experiment that explicitly varies task alignment by tuning the degree of dependence in the synthetic model. As shown in Figure 1(c):
>
> - When the alignment is **strong** $\lambda=0.1$(small $\alpha$), MARL behaves similarly to SARL.
> - When the alignment is **weak** $\lambda=1$ (large $\alpha$), the performance gap widens, with SARL outperforming MARL.
>
> This validates our theoretical prediction regarding the role of $\alpha$.
>
> **2. Synthetic experiments sweeping the number of agents $K$.**
>
> For both independent and dependent subtasks (Figure 1(a) and (b)), we systematically increase $K$. The results show:
>
> - In the **independent** case, increasing $K$ amplifies MARL’s advantage.
> - In the **dependent** case, increasing $K$ magnifies MARL’s inefficiency.
>
> These trends closely follow the predicted structural behavior based on our sample complexity analysis.
>
> **3. A real-world experiment on GSM8K.**
>
> To connect theory with practice, we added a lightweight LLM evaluation using the **Qwen2.5-1.5B-Instruct** model on the **GSM8K** math reasoning dataset.
>
> We designed two MARL/SARL task formulations:
>
> 1. **Independent structure (high alignment):**
>     - We concatenate **two math problems** into a single input.
>     - *Agent 1* solves the first problem; *Agent 2* solves the second.
>     - The two subtasks are independent.
> 2. **Dependent structure (low alignment):**
>     - We decompose each problem into a two-stage pipeline: **(thinking → solving)**.
>     - *Agent 1* generates the chain-of-thought; *Agent 2* produces the final answer.
>     - The correctness of the final output depends heavily on the earlier reasoning, creating strong cross-stage dependence.
>
> For each structure, we compare:
>
> - **SARL:** a single Qwen model handling the entire task end-to-end,
> - **MARL:** two Qwen-based agents, each trained on its respective subtask.
>
> Results presented in Figures 1(d) and 1(e) match our theoretical predictions:
>
> - **Dependent case:** SARL significantly outperforms MARL.
> - **Independent case:** MARL achieves higher accuracy with fewer samples.
>
> This demonstrates that the theoretical insights transfer meaningfully to real LLM-RL reasoning workflows.
>
> Together, these additional experiments provide more comprehensive empirical support for our theoretical findings.
>
> ### **Q2. Code / benchmark release**
>
> Thank you for the suggestion. We will **release the code** used for synthetic empirical study and the new real-life empirical study with GSM8K. The code will be provided via a public repository upon acceptance.
>
> # **Closing**
>
> We thank you again for the helpful comments. The suggested revisions are highly constructive, and we are confident that implementing them will substantially strengthen the paper. We hope our response has satisfactorily addressed your questions, comments and concerns, and can lead to a positive-re-evaluation of the paper. Please let us know if there is any further questions.

---

> > ### Comment · Reviewer_aE3u · 2025-11-26
> >
> > Thank you for the detailed response and clarifications. My main concern has been addressed, and I will increase my score to 6.

---

> > > ### Author Response · Authors · 2025-11-27
> > >
> > > We are very glad to hear that you found our response satisfactory.
> > > Thank you again for your kind consideration and thoughtful feedback — we truly appreciate the time and effort you dedicated to reviewing our work.

---

### Author Response · Authors · 2025-11-24
**Summary of updates in the paper**

#

We sincerely thank all reviewers for their thoughtful comments and constructive suggestions. Based on the feedback, we have revised the paper accordingly. All changes are highlighted in **blue** in the main manuscript for ease of reference. The major revisions include:

1. **Clarified theoretical exposition:**

    Expanded explanations and added missing definitions in the main text to address potential sources of confusion.

2. **Strengthened modeling discussion:**

    Added a more detailed discussion of the modeling assumptions, their scope, and how they relate to real multi-agent LLM systems.

3. **Significantly expanded empirical evaluation:**

    Added new synthetic experiments (including alignment sweeps and agent-count sweeps) and a real-world experiment on GSM8K using Qwen to better demonstrate the practical relevance of the theoretical results.


We greatly appreciate the reviewers’ insights and believe these revisions have substantially improved both the clarity and the impact of the paper.

---

### Author Response · Authors · 2025-11-30
**Executive Summary for AC/SACs (1/2)**

Dear AC/SACs:

We are grateful for the opportunity to submit our work to ICLR and sincerely appreciate the time and effort that you and the reviewers have devoted to the evaluation process. We understand that, in this unfortunate situation, your workload is significantly heavier than usual, and we truly value the attention given to our submission. To help streamline your review, we have prepared the following executive summary, which includes (1) a brief overview of the paper and (2) a concise summary of the reviewers’ questions along with our responses. Our goal is to make it easier for you to navigate the rebuttal and efficiently assess the revisions we have made.

# **Summary of the Paper**

This paper investigates the relative sample efficiency of Multi-Agent Reinforcement Learning (MARL) and Single-Agent Reinforcement Learning (SARL) from a theoretical perspective, addressing conflicting empirical findings in the literature. We derived the sample complexity bounds for both MARL and SARL under two types of task decomposition: **independent** and **dependent** subtasks. Based on these bounds, we explore how the relative performance of MARL and SARL varies with task decomposition and introduce an alignment factor to quantify the effects of task misalignment. These results provide practical insights into designing efficient agentic systems.

Although the primary focus of the paper is theoretical, we also conduct a small empirical study. This includes synthetic experiments with noisy arithmetic regression tasks and a real-world experiment using the Qwen model on the GSM8K dataset. These experiments validate our theoretical predictions regarding the structural relationship between SARL and MARL.

# **Summary of Reviewers' Questions and Our Responses**

We are pleased that all reviewers recognized the timeliness, significance, and rigor of our analysis and results. We are especially grateful that Reviewer **aE3u** engaged with our rebuttal, found our clarifications satisfactory, and subsequently raised their score. Below we summarize the key concerns raised by the reviewers and how we addressed them.

## **Common Questions**

1. **Simplified Empirical Study (aE3u, geeW, oXhZ,osy4)**:
    - **Concern**: The empirical study was too simplified.
    - **Response**: We substantially expanded the empirical section to include a broader set of synthetic experiments (e.g., varying alignment factor and agent count) and a real-world experiment on the GSM8K dataset with QWEN model. These experiments validate our theoretical predictions and improve the paper’s practical relevance.

2. **Simplified Modelling (geeW,oXhZ)**:
   - **Concern**: The multi-agent formulation appeared overly simplified and might not capture the complexity of real LLM-based systems.
   - **Response**: We clarified that the turn-based formulation is an analytical abstraction designed to isolate task dependency—the key factor in MARL vs. SARL efficiency. This abstraction flexibly captures common MARL coordination patterns, including those mentioned by the reviewers. We expanded the manuscript to more clearly articulate the scope and generality of this modelling approach.

---

> ### Author Response · Authors · 2025-11-30
> **Executive Summary for AC/SACs (2/2)**
>
> ## **Reviewer-Specific Concerns**
>
> ### **Reviewer aE3u:**
>
> 1. **Shared Input vs. Partial Observations**:
>     - **Concern**: The reviewer asked about our assumption of shared input versus partial observations and whether this affects the theoretical results.
>     - **Response**: We clarified the observation setup and explained how partial observability would affect only constant factors, not the core sample-complexity results. This clarification has been added to the revised manuscript.
> 2. **Missing Symbol Definitions in Eq. 3.1**:
>     - **Concern**: Several symbols in Eq. 3.1 were not defined clearly.
>     - **Response**: We reinstated the missing definitions and provided a notation table in the appendix to improve clarity.
> 3. **Tightness of the Bound in Theorem 4.2**:
>     - **Concern**: The reviewer questioned the tightness of the bound, particularly the reward dependence.
>     - **Response**: We clarified the reward dependence structure and the origin of the scaling factor, and revised the manuscript accordingly.
> 4. **i.i.d. Samples vs. Correlated RL Trajectories**:
>     - **Concern**: The reviewer asked whether our analysis assumes i.i.d. samples and how this would affect real-world RL environments with temporal correlations.
>     - **Response**: We clarified that the analysis assumes i.i.d. sampling (consistent with common RLHF-style training) and discussed extensions to correlated trajectories. The discussion is included in the manuscript.
> 5. **Summation of Per-Agent Dimensions**:
>     - **Concern**: The reviewer ask whether summing the per-agent dimensions overestimate the complexity when agents share parameters?
>     - **Response**: We clarified that shared components are counted once and explained how our dimensionality notion accounts for parameter sharing.
> 6. **Interpretation and Estimation of the Alignment Factor ($\alpha$)**:
>     - **Concern**: The reviewer asked for a clearer explanation of $\alpha$ and how to estimate $\alpha$.
>     - **Response**: We clarified the interpretation of $\alpha$ and provided detailed procedures for estimating it. These explanations and methods are now included in the revised manuscript.
>
> ### **Reviewer geeW:**
>
> 1. **Results Feel Intuitive**:
>     - **Concern**: The reviewer noted that some of the results might feel intuitive.
>     - **Response**: We clarified that, despite seeming intuitive in hindsight, these results have *not* been intuitive in practice, as demonstrated by contradictory empirical findings in the literature. Our analysis resolves this ambiguity by offering a precise, quantitative, and actionable framework that characterizes the key factors governing the relative efficiency of MARL versus SARL.
>
> ### **Reviewer oXhZ:**
>
> 1. **Theoretical and Practical Connection of Hyperparameters**:
>     - **Reviewer Concern**: The reviewer asked how the theoretical bounds map to practical LLM training settings.
>     - **Response**: We provided a detailed mapping between the theoretical quantities used in our analysis and practical LLM components. This mapping is also referenced and included in the appendix of revised manuscript.
>
> ### **Reviewer osy4:**
>
> 1. **Potential Gap in Using Covering Number Bounds**
>     - **Concern**: The reviewer expressed concern about the use of covering-number bounds for deriving the sample complexity.
>     - **Response**: We clarified and discussed why covering-number bounds are appropriate for our comparative analysis. We also emphasized that the empirical results support the qualitative trends predicted by these theoretical bounds. The clarification and discussion are added to the appendix of the revised manuscript.
>
> ## **Closing**
>
> This summary outlines how we have thoroughly addressed all reviewer concerns and made significant revisions to strengthen the paper. We would once again like to thank all reviewers for their thoughtful questions, comments, and suggestions. The added experiments, expanded discussions, and clarifications improve the paper’s clarity, rigor, and practical relevance.

---

### Meta-Review · Area_Chair_A876 · 2026-01-05

**Summary:**

This paper presents a theoretical analysis comparing the sample efficiency of Multi-Agent Reinforcement Learning (MARL) and Single-Agent RL (SARL) in the context of LLM-based agentic systems. Using the PAC framework, the authors derive sample complexity bounds under independent and dependent task decompositions and introduce the concept of task alignment to quantify the impact of decomposition mismatch. The empirical validation, while improved, remains limited in scale and realism compared to full LLM training pipelines.

**Reviewer Concerns:**

For reviewer geeW, the primary criticism was regarding perceived intuitive results, which the authors contextualized effectively. For reviewers  aE3u, oXhZ, and osy4, the concerns are about missing symbol definitions and an expanded discussion on modeling assumptions and their relevance to real multi-agent LLM systems.

**Reviewer Scores:**

The overall score is marginally below the acceptance threshold.

---

### Decision · Program_Chairs · 2026-01-26

Reject